# Regulation of the EphA2 receptor intracellular region by phosphomimetic negative charges in the kinase-SAM linker

Bernhard C. Lechtenberg ⬡ [1,2,3✉], Marina P. Gehring[1], Taylor P. Light ⬡ [4], Christopher R. Horne ⬡ [2,3], Mike W. Matsumoto[1], Kalina Hristova ⬡ [4] & Elena B. Pasquale ⬡ [1✉]

Eph receptor tyrosine kinases play a key role in cell-cell communication. Lack of structural information on the entire multi-domain intracellular region of any Eph receptor has hindered understanding of their signaling mechanisms. Here, we use integrative structural biology to investigate the structure and dynamics of the EphA2 intracellular region. EphA2 promotes cancer malignancy through a poorly understood non-canonical form of signaling involving serine/threonine phosphorylation of the linker connecting its kinase and SAM domains. We show that accumulation of multiple linker negative charges, mimicking phosphorylation, induces cooperative changes in the EphA2 intracellular region from more closed to more extended conformations and perturbs the EphA2 juxtamembrane segment and kinase domain. In cells, linker negative charges promote EphA2 oligomerization. We also identify multiple kinases catalyzing linker phosphorylation. Our findings suggest multiple effects of linker phosphorylation on EphA2 signaling and imply that coordination of different kinases is necessary to promote EphA2 non-canonical signaling.

[1] Cancer Center, Sanford Burnham Prebys Medical Discovery Institute, La Jolla, CA 92037, USA. [2] Ubiquitin Signalling Division, The Walter and Eliza Hall Institute of Medical Research, Parkville, VIC 3052, Australia. [3] Department of Medical Biology, The University of Melbourne, Parkville, VIC 3010, Australia. [4] Department of Materials Science and Engineering, Institute for NanoBioTechnology, Johns Hopkins University, 3400 Charles Street, Baltimore, MD 21218, USA. ✉email: lechtenberg.b@wehi.edu.au; elenap@sbpdiscovery.org

The Eph receptors are the largest receptor tyrosine kinase family, and together with their cell surface-anchored ephrin ligands represent an important cell-cell communication system that regulates a multitude of physiological and pathological processes[1–4]. The fourteen Eph receptors have a conserved domain structure, with an extracellular portion including the N-terminal ligand-binding domain and several other domains[5,6]. The intracellular portion of the Eph receptors includes a juxtamembrane segment, the tyrosine kinase domain, a linker, a sterile alpha motif (SAM) domain, and a short C-terminal tail containing a PDZ domain-binding motif (Fig. 1a). The juxtamembrane segment and activation loop in the kinase domain control kinase activity through a mechanism involving tyrosine phosphorylation[7–9]. Depending on the Eph receptor, the SAM domain can also differentially affect kinase activity[10–13]. How the SAM domain affects kinase activity and the potential interplay between the juxtamembrane segment, kinase domain and SAM domain are poorly understood[14]. While structural information on the extracellular portion of Eph receptors has provided important insights[5,15], structural information on the entire intracellular region is not available.

EphA2 has been implicated in many physiological and pathological processes, including epithelial homeostasis, immune system function, angiogenesis, inflammation, atherosclerosis, parasitic infections, and cancer malignancy[2–4,16]. EphA2 regulates these diverse processes by signaling through different mechanisms. EphA2 canonical signaling is induced by ephrin ligand binding and is mediated by kinase activity and autophosphorylation on tyrosine residues. In tumor endothelial cells, EphA2 canonical signaling promotes tumor angiogenesis[2,17–19]. In cancer cells, EphA2 tyrosine phosphorylation is often low, consistent with the unusual effects of EphA2 canonical signaling, which inhibits major oncogenic networks such as RAS-ERK and AKT-mTORC1[2,20,21]. EphA2 can also signal through a completely different, more recently discovered non-canonical signaling mechanism that involves phosphorylation on serine 897 (S897). EphA2 non-canonical signaling is often elevated in cancer cells and promotes their oncogenic properties, including epithelial-mesenchymal transition, invasiveness and metastasis, mechanotransduction, stem cell-like properties, and drug resistance[2,3,22–25]. Despite its profound significance in cancer, the mechanism of non-canonical signaling is not well understood[22,26].

It is unknown how phosphorylation of S897 in the linker connecting the EphA2 kinase and SAM domains regulates EphA2 signaling. The kinase-SAM linker contains a cluster of five serine/threonine residues that besides S897 include S892, T898, S899 and S901 (Fig. 1a). These residues can also be phosphorylated[27,28] (phosphosite.org), but it remains unclear

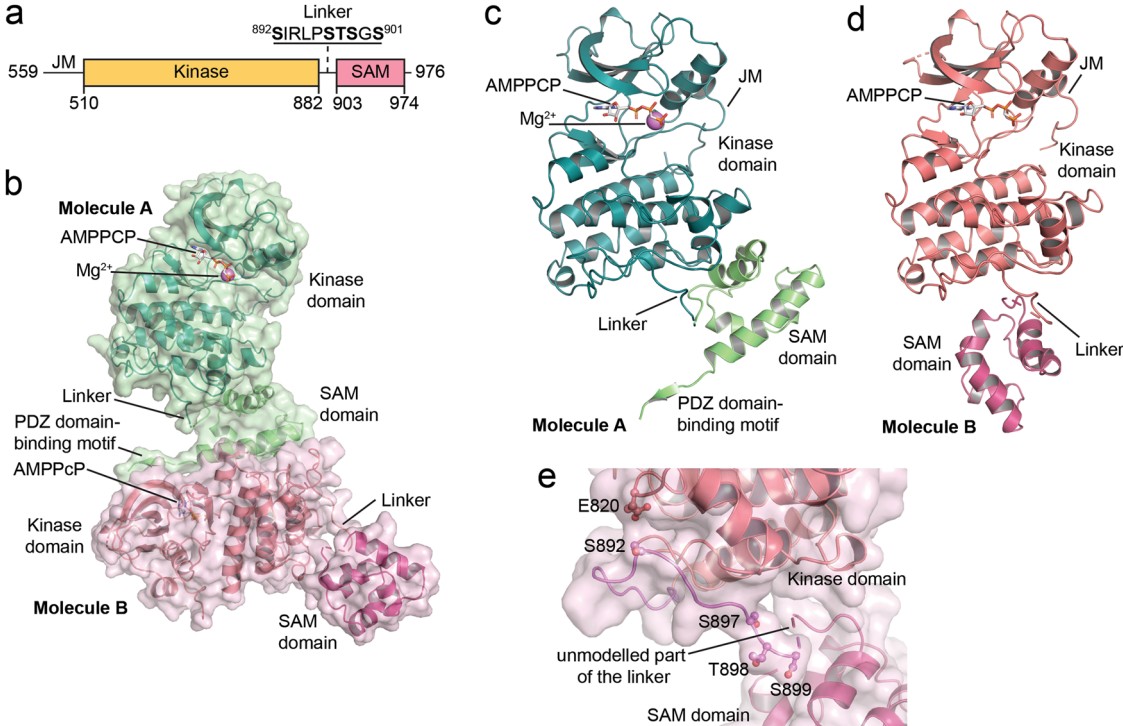

**Fig. 1 Crystal structure of the EphA2 wild-type intracellular region. a** Schematic illustrating the domains of the EphA2 intracellular region. **b** Overview of the two EphA2 molecules in the crystallographic asymmetric unit; resolution 1.75 Å. Molecule A is shown in green, with the kinase domain in darker green and the SAM domain and C-terminal tail in lighter green. The kinase and SAM domains of molecule B are shown in two different shades of red. The ATP-analog β,γ-methyleneadenosine 5′-triphosphate (AMPPCP), present in the active sites of both molecules A and B, is shown as gray sticks and a Mg²⁺-ion in the active site of molecule A is shown as a purple sphere. The kinase domains in both molecules adopt the active DFG-in conformation. The regions that are well defined in both EphA2 molecules include the portion of the juxtamembrane segment encoded by our construct, most of the kinase domain, and the whole SAM domain. Parts of the activation loop (L760-I779), residues S636-G637 in the short β1-2 loop of the N-lobe of EphA2 molecule B, and parts of the kinase-SAM linker (S899-V904 in molecule A and G900-V904 in molecule B) are not defined due to lack of electron density. The C-terminal tail (K966-I976) including the PDZ domain-binding motif is defined only in molecule A. **c** Rotated view of molecule A, highlighting the compact arrangement of the kinase and SAM domains. Missing portions of the linker in **c** and **d** are shown as a dashed line. **d** Rotated view of molecule B, with the kinase domain in the same orientation as for molecule A in panel **c**. The kinase and SAM domains are in a less compact arrangement. **e** Detail of the kinase-SAM linker of molecule B, showing that the S892, S897, T898, and S899 phosphorylation sites are solvent exposed and accessible. The fifth phosphorylation site (S901) is in the undefined region of the linker, which is indicated by a dashed connector. E820 is in close proximity (3–5 Å) to S892 and might be an allosteric sensor of linker phosphorylation.

whether their phosphorylation plays a role in EphA2 non-canonical signaling. The significance of other serine/threonine phosphorylation sites identified in the EphA2 juxtamembrane segment and kinase domain (phosphosite.org) is even less clear.

In this work, we investigate the interplay of the different domains and linkers in the EphA2 intracellular region and mechanistically elucidate how phosphorylation of the kinase-SAM linker affects EphA2 function. Recombinant expression of most of the EphA2 intracellular region enables us to solve its crystal structure and investigate its dynamic behavior in solution. Our findings support a model in which multiple negative charges introduced by linker phosphorylation cooperate to promote an ensemble of open conformations of the EphA2 intracellular region that mediate non-canonical signaling. We find that multiple EphA2 kinase-SAM linker residues can be simultaneously phosphorylated by different kinases, consistent with the notion that cumulative linker phosphorylation is important for EphA2 non-canonical signaling.

## Results

**S897 is part of a cluster of phosphosites in the EphA2 kinase-SAM linker**. Mass spectrometry data from the PhosphoSitePlus database show that all five serine/threonine residues in the EphA2 kinase-SAM linker are frequently phosphorylated in cell lines and tissues (Supplementary Fig. 1a). By screening multiple cancer cell lines in which EphA2 is known to be phosphorylated on S897[27,29], we confirmed phosphorylation on two other residues in the linker, S892 and S901 (Supplementary Fig. 1b, c). We observed a strong correlation in the extent of EphA2 phosphorylation on S897 and S901 (Supplementary Fig. 1d), consistent with our previous findings that EphA2 S897 phosphorylation primes S901 phosphorylation[28]. Analysis of published mass spectrometry data profiling 27 non-small cell lung cancer and 4 breast cancer cell lines[29] shows a very strong correlation of EphA2 S897 phosphorylation with phosphorylation on T898, S899 and S901 and a weaker correlation with S892 phosphorylation (Supplementary Fig. 1e). Accordingly, mass spectrometry data show that more than 60% of the EphA2 kinase-SAM linker peptides analyzed are phosphorylated on multiple residues simultaneously (Supplementary Fig. 1f)[27,30–35], with up to four phosphorylation sites detected in a single peptide[35] and many different combinations of phosphorylation sites (Supplementary Fig. 1g). Thus, EphA2 kinase-SAM linker phosphorylation is highly cooperative, and S897 is typically phosphorylated together with other linker residues.

**Structure of the EphA2 intracellular region**. To obtain insights into EphA2 signaling mechanisms, we solved the crystal structure of most of the unphosphorylated EphA2 intracellular region (residues D590-I976; Fig. 1b–d, Supplementary Table 1). The crystallographic asymmetric unit contains two EphA2 molecules (molecules A and B; Fig. 1b–d), even though the EphA2 intracellular region is monomeric in solution (Supplementary Fig. 2). Interestingly, residues D590-A600 in the juxtamembrane segment adopt a different conformation compared to previous structures of EphA2 and other Eph receptors containing the juxtamembrane segment and kinase domain. In our structure, the αA′-helix is unwound and Y594 (the second of two conserved juxtamembrane tyrosine residues) occupies a pocket previously found to be occupied by the first conserved tyrosine (Y588 in EphA2, which is not part of our construct) (Supplementary Fig. 3a, b)[9,36], indicating a dynamic conformation of the juxtamembrane segment even when its tyrosine residues are unphosphorylated.

The relative orientation of the kinase and SAM domains is different in the two EphA2 molecules, suggesting that the EphA2 intracellular region can assume different conformations, likely

enabled by flexibility in the kinase-SAM linker. The N-terminal part of the linker (up to residue T898) and the beginning of the SAM domain (starting at F906) adopt the same conformations in both molecules (Supplementary Fig. 3c, d). In contrast, residues S899-P905 (including the S899 and S901 phosphorylation sites) are mostly undefined in both EphA2 molecules (Fig. 1b–d), likely due to high conformational flexibility. Thus, conformational differences in these residues appear to be the main reason for the different relative orientations of the kinase and SAM domains in the two EphA2 molecules.

In both EphA2 molecules, the N-terminal part of the linker is partly wrapped around the bottom of the kinase domain C-lobe (Fig. 1e) and locked in place by interactions of I893 with a hydrophobic pocket in the kinase domain (Supplementary Fig. 3c). This linker conformation is different from that observed in EphA2 structures that do not contain the SAM domain, where most of the linker (including I893) faces away from the kinase domain (Supplementary Fig. 3c), suggesting that the SAM domain affects linker conformation. Notably, the conformation of the N-terminal part of the EphA2 linker and the interaction of I893 with a pocket in the kinase domain resemble features observed in an EphA3 structure lacking the SAM domain, while the central parts of the EphA2 and EphA3 linkers have distinct conformations (Supplementary Fig. 3c)[37].

**Linker phosphomimetic mutations regulate kinase-SAM arrangement**. To determine whether the negative charges introduced by phosphorylation of the linker might affect the kinase-SAM domain arrangement, we solved the structures of two EphA2 phosphomimetic mutants. In the first, we replaced both S897 and S901 with glutamic acid, since glutamate is known to at least in part functionally mimic a phosphorylated residue when the effects of phosphorylation are due to the negative charge of phosphate[38,39]. The structure of the EphA2 S897E/S901E double mutant shows a different kinase-SAM domain arrangement than EphA2 WT (Fig. 2, Supplementary Table 1). In the single molecule of the structure, the SAM domain is completely detached from the kinase domain and the C-terminal part of the linker is fully extended (Fig. 2a). We did not observe any substantial differences in the overall fold of the individual domains between the WT and S897E/S901E structures, and all EphA2 molecules in both structures adopt the active DFG-in conformation. Unlike the EphA2 WT structure, the activation loop is fully resolved in the S897E/S901E mutant structure, potentially due to stabilizing crystal contacts. In the S897E/S901E structure, however, the central portion of the kinase-SAM linker adopts a conformation different from WT and appears to be stabilized by a salt bridge between E897 (replacing S897) and R890 (Fig. 2b). Additionally, a key difference is evident in the αFG loop of the kinase domain, including W819 and adjacent residues (Fig. 2b), a region previously identified in EphA3 as part of an allosteric network connecting the juxtamembrane segment, activation loop and kinase-SAM linker[14].

In the second EphA2 mutant, we replaced only S901 with glutamic acid. The EphA2 S901E structure is essentially identical to the WT structure (Supplementary Fig. 4; Supplementary Table 1), in agreement with the lack of definition of residue S901 in the structures and suggesting that S901 phosphorylation alone does not significantly affect the conformation of the EphA2 intracellular region.

These data suggest that linker negative charges regulate the arrangement of the EphA2 kinase and SAM domains and may induce conformational changes in the αFG loop of the kinase domain. However, since the kinase and SAM domains arrangement in the various crystal structures might be differentially

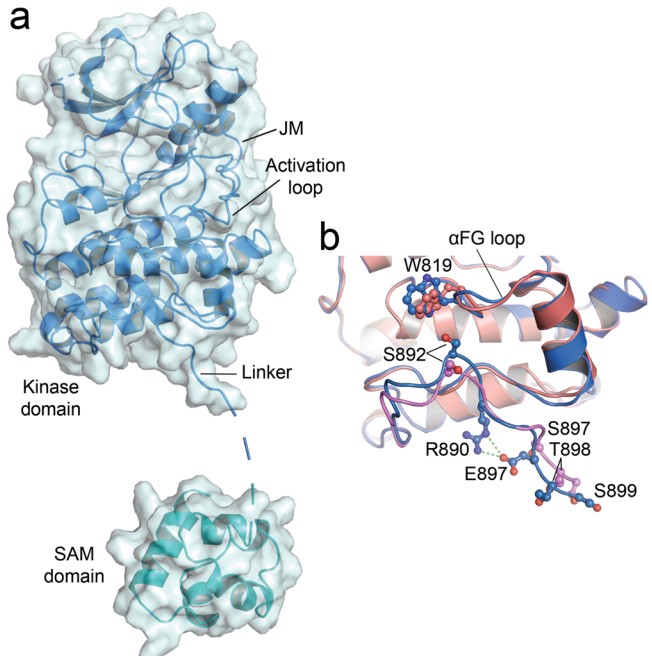

**Fig. 2 Crystal structure of the intracellular region of the EphA2 S897E/ S901E phosphomimetic mutant. a** Overview of the asymmetric unit containing a single EphA2 molecule in an elongated conformation; resolution 2.8 Å. The kinase domain is shown in blue and the SAM domain in cyan. The structure includes most of the juxtamembrane segment (JM) and the kinase domain, the full activation loop (including residues L760-I779, which are not defined in the WT structure), and the SAM domain (residues V909-L965). Parts of the linker region and N-terminus of the SAM domain (residues 900–908, dashed line) and the C-terminal tail (K966-I976) are not visible in the structure due to missing electron density. **b** Comparison of the resolved linker regions in the structures of EphA2 WT molecule B (colored as in Fig. 1d) and the S897E/S901E mutant (blue), highlighting key differences in the linker structures. Key residues are shown as sticks and labeled. R890 and the phosphomimetic E897 in the S897E/ S901E structure form a salt bridge (green dashes). The SAM domains are omitted for clarity.

affected by crystal packing, we used size-exclusion chromatography in line with small angle X-ray scattering (SEC-SAXS) to obtain information on the EphA2 intracellular region in solution.

**Linker phosphomimetic mutations elongate the intracellular region**. We determined the low-resolution solution structure of the entire unphosphorylated EphA2 intracellular region, except for the first 10 residues (S570-I976). This portion of EphA2 comprises 20 additional juxtamembrane residues compared to the portion analyzed by X-ray crystallography and includes both conserved Y588 and Y594 residues, which interact with the kinase domain in unphosphorylated Eph receptors[9]. We compared EphA2 WT and four mutants that mimic different kinase-SAM linker phosphorylation states: S892E, S897E, 3E (T898E/S899E/ S901E) and 5E (S892E/S897E/T898E/S899E/S901E; Fig. 3a, Supplementary Table 2). Guinier analysis yielded a linear plot, consistent with a single monodisperse species in solution and an absence of substantive aggregation or interparticle interference (Fig. 3b). The SAXS curves for the S892E, S897E and 3E mutants largely overlap with the WT curve (Fig. 3a), suggesting that the conformations of these EphA2 mutants are similar to WT in solution. However, the SAXS curve for the EphA2 5E mutant diverges from the others, especially in the low $q$ region (<0.15) (Fig. 3a, inset). Additionally, the SAXS pairwise distance

distribution plot shows a peak with a prominent tail towards longer distances for EphA2 WT, indicative of an elongated molecule, with a maximum particle dimension ($D_{max}$) of 104 Å (Fig. 3c, Supplementary Table 2). The S897E and 3E mutants show comparable distance distributions to EphA2 WT, with the same $D_{max}$ of 104 Å. Interestingly, EphA2 5E ($D_{max}$ = 123 Å) and to a lesser extent EphA2 S892E ($D_{max}$ = 116 Å) feature a broader distance distribution with a more prominent tail skewed towards longer distances, indicating that negative charges in the kinase-SAM linker induce a conformational change leading to a more elongated EphA2 intracellular region.

Ab-initio model building from the EphA2 WT and 5E SAXS data yields elongated envelopes with a bulky top and a narrower bottom (Fig. 3d). The EphA2 5E envelope is more extended than the WT envelope, as expected from the differences in the distance distribution plots (Fig. 3c, d). Overlay of the SAXS envelopes with our EphA2 crystal structures indicates that the WT envelope best matches EphA2 molecule B (Fig. 3e). The EphA2 5E envelope best fits the more elongated conformation of the EphA2 S897E/ S901E crystal structure, but in general shows less agreement with the crystal structures (Fig. 3e), suggesting that the EphA2 intracellular region adopts an ensemble of conformations in solution that is shifted towards more elongated conformations by phosphomimetic negative charges in the linker.

**Hydrogen-deuterium exchange supports kinase SAM domain interaction**. SAXS provides information about the conformation of a protein in solution but, on its own, yields limited detail on protein dynamics. Thus, to further investigate the dynamics of the EphA2 intracellular region, we used hydrogen-deuterium exchange in combination with mass spectrometry (HDX-MS) to analyze the same unphosphorylated EphA2 WT intracellular portion used for SAXS. We measured HDX at different time points up to 5 min (Fig. 4, Supplementary Figs. 5, 6). This time frame identifies regions in which protein backbone protons that are solvent exposed, and not engaged in stable hydrogen bonds, rapidly exchange with deuterons from the solvent[40,41]. We found that the core of the kinase domain, including the C-terminal half of the αC-helix, the hydrophobic spines and the αF-helix, undergo limited exchange within the 5 min time course, indicating that these regions are shielded from the solvent, as expected (Fig. 4a). Conversely, the juxtamembrane segment, Gly-rich loop, activation loop, and kinase-SAM linker show rapid exchange even at the earliest time point (0.5 min; Fig. 4a), indicating that these regions are solvent exposed and highly dynamic (Fig. 4b), in agreement with the crystal structures. The solvent exposed, unstructured conformation of the kinase-SAM linker is typical of protein segments containing multiple phosphorylation sites[42]. The high solvent exposure of the juxtamembrane segment suggests that the binding of the conserved juxtamembrane tyrosine residues to pockets in the kinase domain is not sufficiently stable to detectably alter HDX, consistent with the fact that these tyrosine residues are phosphorylation sites and thus must be accessible. The observed dynamics of the juxtamembrane segment are also consistent with the discrepancies observed in different crystal structures, which show that Y588 and Y594 can both interact with the same pocket of the EphA2 kinase domain (Supplementary Fig. 3a, b).

In general, the extent of exchange is similar at the different time points examined (0.5 to 5 min). However, in two regions we observed intermediate exchange that gradually increases from 0.5 to 5 min (Fig. 4a, b). One of these regions corresponds to residues P780-M850, comprising the αF-αH helices in the C-lobe of the kinase domain (Fig. 4b). Parts of this region form a prominent negatively charged surface (Fig. 4c). The other region is the

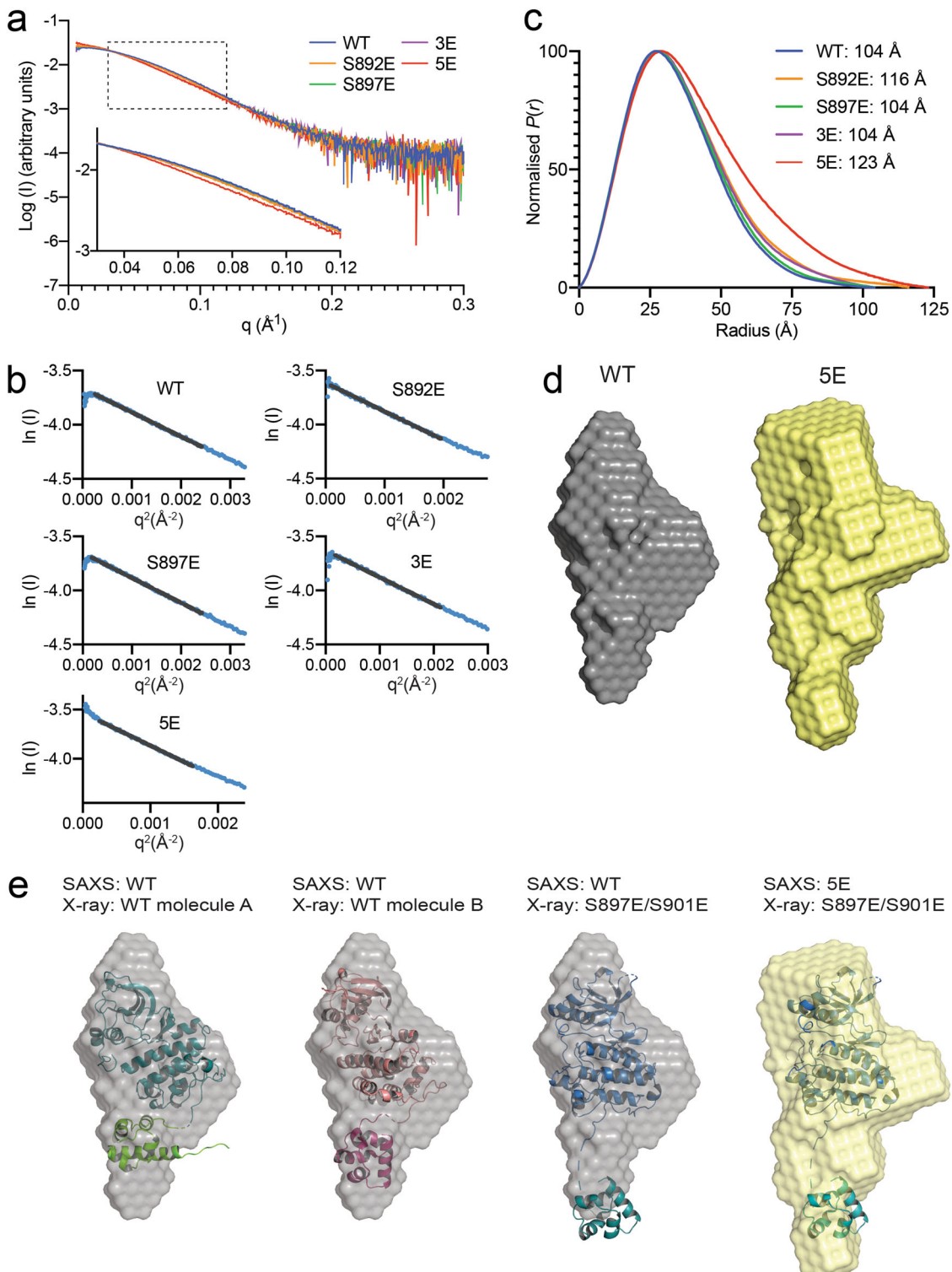

**Fig. 3 SEC-SAXS analysis of the EphA2 intracellular region. a** Small angle X-ray scattering profiles of averaged, and background subtracted data from the apex of the inline size exclusion chromatography peak for EphA2 WT (residues S570-I976) and 5 mutants as indicated. **b** Guinier analysis of the data shown in **a**; the Guinier fit is indicated by a black line. **c** Pairwise distance distribution, $P(r)$, plot, calculated from scattering data with GNOM. Maximum particle dimensions ($D_{max}$) are indicated for each mutant. **d** Ab initio SAXS envelopes for EphA2 WT and EphA2 5E calculated with DAMIFF. **e** Comparison between SAXS envelopes for EphA2 WT and 5E mutant with the crystal structures of EphA2 WT (molecules A and B) and the S897E/S901E mutant. Source data for **a**, **b**, **c** are provided in the Source Data file.

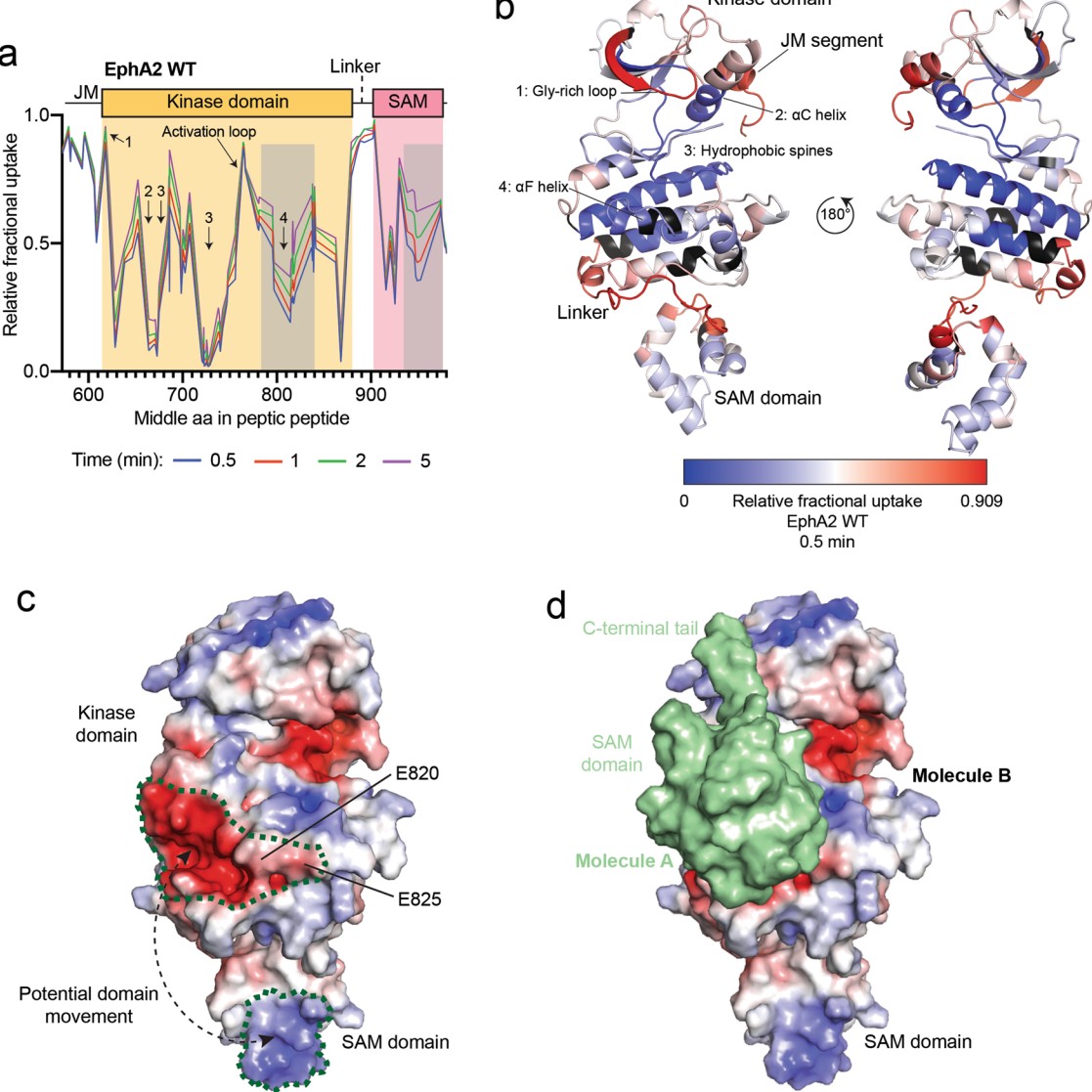

**Fig. 4 HDX-MS analysis of the EphA2 WT intracellular region. a** Relative fractional deuterium uptake plot for peptic peptides identified from EphA2 WT after 0.5, 1, 2 and 5 min. Key domains of the EphA2 intracellular region are indicated and parts of the kinase domain are indicated with Arabic numerals (see also panel **b**). The gray shaded areas indicate regions of intermediate exchange, as described in the text. **b** Relative fractional deuterium uptake after 0.5 min, mapped onto the EphA2 WT structure (molecule B). Residues without coverage are colored in black. Domains and key regions are indicated. **c** Electrostatic surface of the EphA2 WT structure in the same orientation as in **b**, left. The positively-charged (blue) region in the SAM domain and the negatively-charged (red) region in the kinase domain are outlined with a green dotted line. A potential domain movement is indicated by dashed line with arrows. This interaction is possible due to the 20-amino acid flexible linker connecting the SAM and kinase domains. **d** Intermolecular interaction between the EphA2 kinase and SAM domains as observed in our EphA2 WT crystal structure. EphA2 molecule B is shown as in panel **c** with the SAM domain of molecule A shown as green surface. Source data for **a** are provided in the Source Data file.

C-terminal portion of the SAM domain (residues Y930-N970), which forms a positively charged surface (Fig. 4c). The correlation of the two exchange processes and the complementary charges of the two surfaces suggest that the kinase and SAM domains interact with each other, shielding the interacting surfaces from the solvent (Fig. 4b, c). This hypothesis is further supported by our crystal structure of EphA2 WT, where the SAM domain of molecule A interacts with the αF–αH helices in the kinase domain of molecule B (Figs. 1b, 4d), which may mimic a physiologically relevant intramolecular interaction.

**Negative charges in the kinase-SAM linker alter EphA2 conformation.** Considering our SAXS data, we hypothesized that

linker phosphorylation may lead to effects whose magnitude depends on the number of residues simultaneously phosphorylated. We therefore analyzed the EphA2 5E mutant, in which all five linker phosphosites are replaced by glutamic acid, to maximize our ability to detect potential effects of linker phosphorylation. The most drastic effect in this mutant, compared to EphA2 WT, is increased HDX in the C-lobe of the kinase domain and most of the SAM domain (Fig. 5a, b, Supplementary Figs. 5, 6), suggesting that negative charges in the kinase-SAM linker disrupt the intramolecular interaction between the two domains. Given these results, we compared the HDX kinetics for EphA2 WT and the four phosphomimetic mutants we previously analyzed by SAXS: 5E, 3E, S892E and S897E (Fig. 5, Supplementary Figs. 5, 6).

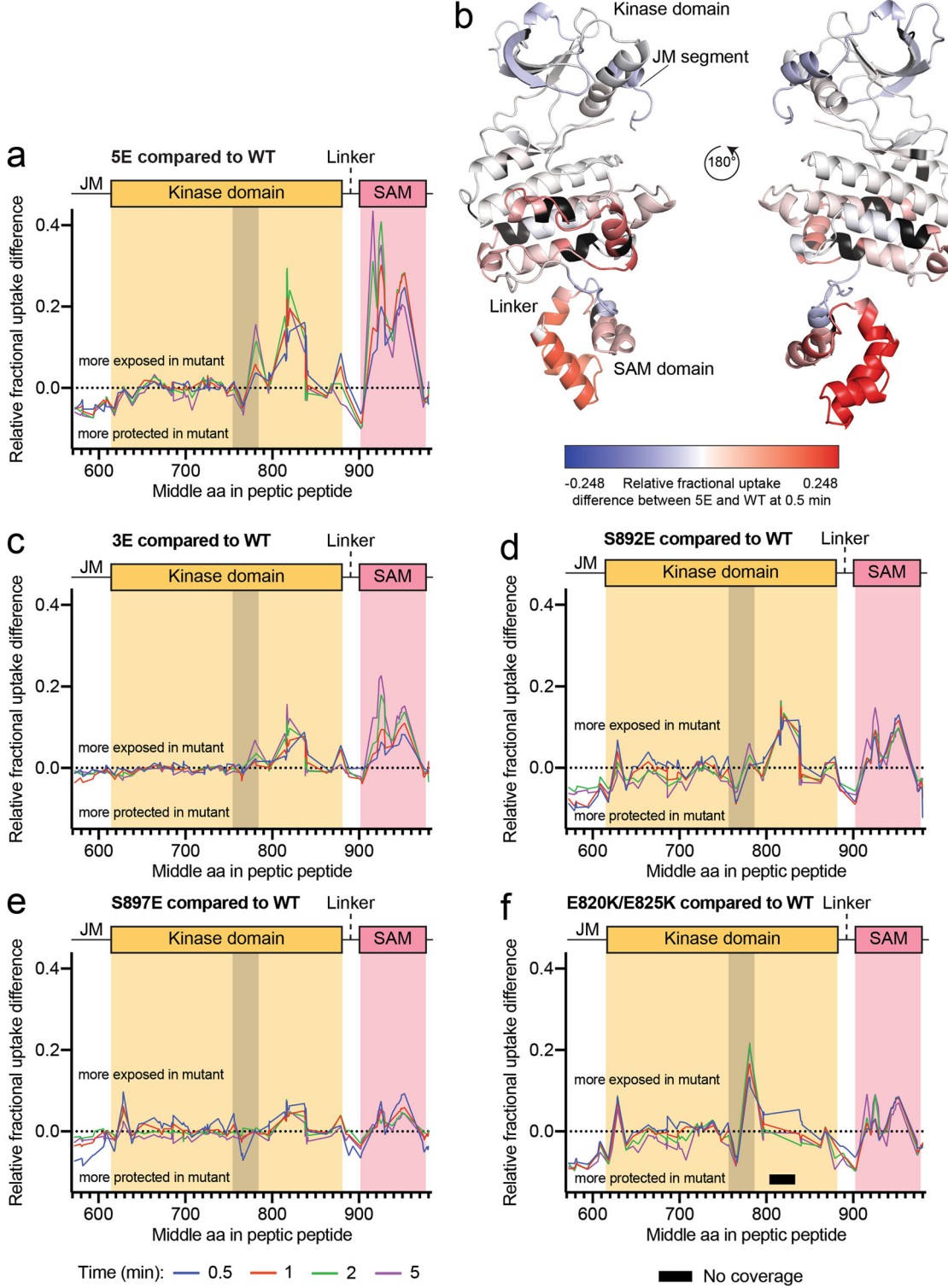

**Fig. 5 HDX-MS analysis of the EphA2 intracellular region with kinase-SAM linker phosphomimetic mutations. a** Fractional deuterium uptake difference plot comparing the EphA2 5E mutant with EphA2 WT after 0.5, 1, 2 and 5 min of exchange. Positive values indicate more rapid exchange in the 5E mutant compared to WT, whereas negative values indicate slower exchange in the mutant compared to WT. The gray shaded area indicates the activation loop (D757-E787). **b** Relative fractional deuterium uptake differences between WT and the 5E mutant after 0.5 min mapped onto the EphA2 WT structure (molecule B). Relative fractional differences from −0.248 (less exposed in the 5E mutant) to 0.248 (more exposed in the 5E mutant) are colored in a gradient from blue to red. Residues without coverage are colored in black. **c–f** Graphs as in panel **a**, comparing fractional deuterium uptake differences between the indicated EphA2 mutants and EphA2 WT. The gray shaded area indicates the activation loop (D757-E787) and the black bar in **f** indicates a region without peptide coverage in the E820K/E825K mutant. Source data for **a**, **c**, **d**, **e**, **f** are provided in the Source Data file.

Of all the mutants, EphA2 5E shows the most accelerated exchange compared to EphA2 WT, especially in the C-lobe of the kinase domain and the SAM domain (Fig. 5a, b). Surprisingly, EphA2 S897E, with mutation of the functionally best characterized linker phosphorylation site, shows the weakest HDX differences in these regions (Fig. 5e), suggesting that phosphorylation of S897 alone is not sufficient to induce EphA2 conformational changes that depend on linker negative charges. A negative charge at position 892 has a stronger effect on HDX than S897 phosphorylation (Fig. 5d), in agreement with the SAXS data (Fig. 3c). The three glutamic acid residues at positions corresponding to the other three phosphosites in the 3E mutant also accelerate exchange slightly more than the individual S897E mutation (Fig. 5c), suggesting differential effects of the various phosphosites.

We also examined the E820K/E825K mutant, in which we replaced two glutamic acid residues in the negatively charged surface of the kinase domain C-lobe with positively-charged lysine residues (Fig. 5f, Supplementary Figs. 5, 6), which we hypothesized would disrupt the observed intramolecular interaction even in the absence of linker phosphorylation. The EphA2 E820K/E825K mutant shows increased exchange in the SAM domain and in the region of the kinase domain C-lobe that precedes the introduced mutations (residues P780-V800) (Fig. 5f). Although we did not detect peptides covering residues I805-G833 for this mutant, these observations are consistent with the involvement of the electronegative surface of the EphA2 kinase domain in the interaction with the SAM domain (Fig. 5f).

Interestingly, in the EphA2 5E, E820K/E825K and S892E mutants, and to a smaller extent in the S897E mutant, we observed a strong correlation in the stabilization of kinase-SAM linker, juxtamembrane segment and parts of the activation loop (L760-A770), all of which undergo lower HDX (Fig. 5a, d–f). This may suggest a regulatory network connecting these regions that critically depends on E820 in the αFG loop, which is close to S892 (Fig. 1e). The decreased HDX of the linker in these mutants suggests that the negatively-charged linker may be engaged in interactions with one of the domains rather than being fully flexible and disordered. In contrast, we did not observe these effects in the 3E mutant, in which HDX in these regions is similar to EphA2 WT (Fig. 5c). A parsimonious interpretation of the data suggests a mechanism where phosphorylation of the N-terminal portion of the linker (mostly S892, but also S897) initiates changes within the linker and has long-range conformational effects on the juxtamembrane segment and the activation loop, whereas the overall level of linker phosphorylation controls the interactions between the kinase and SAM domains, shifting the equilibrium from more closed to more open conformations.

**Linker negative charges affect EphA2 oligomerization.** The proposed intramolecular interaction of the SAM and kinase domains in EphA2 molecules that are positioned side-by-side in the plasma membrane could inhibit their dimerization, possibly by preventing intermolecular interactions between the negatively-charged surface of the kinase domain (occupied by the SAM domain) and a positively-charged surface on the opposite side of the kinase domain (Supplementary Fig. 7). We therefore used fully quantified spectral imaging Förster Resonance Energy Transfer (FSI-FRET)[43] to examine how EphA2 mutations mimicking or preventing phosphorylation of kinase-SAM linker residues affect EphA2 dimerization in HEK293 cells in the absence of ligand stimulation. To obtain dimerization curves, FRET efficiencies (which report on the interaction strength between receptor molecules) were measured along with the concentration of transiently transfected EphA2 mutants labeled

with mTurquoise or EYFP (a FRET pair) (Supplementary Fig. 8a). Curve fitting yielded the dissociation constants $K_{diss}$ for the EphA2 mutants, which were compared to the $K_{diss}$ value previously determined for EphA2 WT[44]. This showed that the phosphomimetic mutations in the linker have negligible effect on EphA2 dimer stability compared to EphA2 WT (in which the linker is phosphorylated[28]), while mutations to the non-phosphorylatable alanine appear to have a small destabilizing effect (Supplementary Fig. 8b).

Since it is well established that the ephrinA1-Fc ligand causes the formation of large EphA2 oligomers that can be visualized by fluorescence microscopy[5,6], we also imaged the plasma membrane at the bottom of HEK293 cells expressing the EphA2 5A (S892A/S897A/T898A/S899A/S901A) or 5E mutant tagged with EYFP, with or without ephrinA1-Fc stimulation (Fig. 6a). EphrinA1-Fc induced a more evident increase in fluorescent patches for EphA2 5E than for EphA2 5A, suggesting that the non-phosphorylated linker of the EphA2 5A mutant does not support the formation of higher-order oligomers.

Since FRET analyses cannot discern oligomer size for oligomers larger than dimers or provide information about heterogeneous populations of oligomers[45], we analyzed the EphA2 oligomers using the fluorescence intensity fluctuation (FIF) technique, which calculates molecular brightness in small sections of the plasma membrane and creates a histogram of the molecular brightness values[46]. This revealed that in the absence of ligand treatment the brightness distribution curves for the EphA2 5A and 5E mutants are between the curves for the monomer control (LAT) and the dimer control (E-cadherin)[47], suggesting the presence of a mixture of monomers and dimers for both EphA2 mutants (Fig. 6b). Furthermore, in agreement with the FRET data, the curve for EphA2 5E is slightly closer to the E-cadherin curve, suggesting higher dimerization of the EphA2 5E than the 5A mutant. Treatment with ephrinA1-Fc only slightly shifts the brightness distribution curve for EphA2 5A (Fig. 6c, d), whereas the curve for EphA2 5E is markedly shifted to brightness values higher than for E-cadherin (Fig. 6c, e), indicating the formation of oligomers larger than dimers, in agreement with the observed fluorescence patches (Fig. 6a). Thus, the EphA2 5E mutant, but not the 5A mutant, forms higher-order oligomers in response to ligand stimulation, as previously described for EphA2 WT[5,6]. Taken together, our FRET and FIF analyses suggest that kinase-SAM linker phosphorylation promotes EphA2 oligomerization in the plasma membrane.

**Multiple kinase families can phosphorylate the kinase-SAM linker.** Previous studies have shown that the AGC kinases AKT, RSK and PKA phosphorylate EphA2 on S897[22,27,33], CK1 family kinases phosphorylate S901[27] and PKC family kinases phosphorylate S892[28]. To systematically identify the spectrum of kinases that phosphorylate the EphA2 kinase-SAM linker, we screened a collection of 298 kinases representing most of the serine/threonine kinase families using an in vitro kinase reaction. As the substrate, we used a peptide corresponding to EphA2 residues D886–T908. In this peptide, S897 was already phosphorylated, to preclude phosphorylation by kinases targeting this site. This assay revealed that multiple kinases readily phosphorylate the peptide substrate, with 15% (i.e. 45) of the kinases tested accounting for ~70% of the total radioactivity incorporated (Fig. 7a; Supplementary Table 3). Members of several kinase families caused high peptide phosphorylation (>50,000 cpm; Fig. 7b; Supplementary Table 3), including nearly all PKC family members, PAK3, and members of the CK1 and NEK families.

We previously reported that constitutively active acidophilic CK1 family kinases phosphorylate EphA2 on S901 following

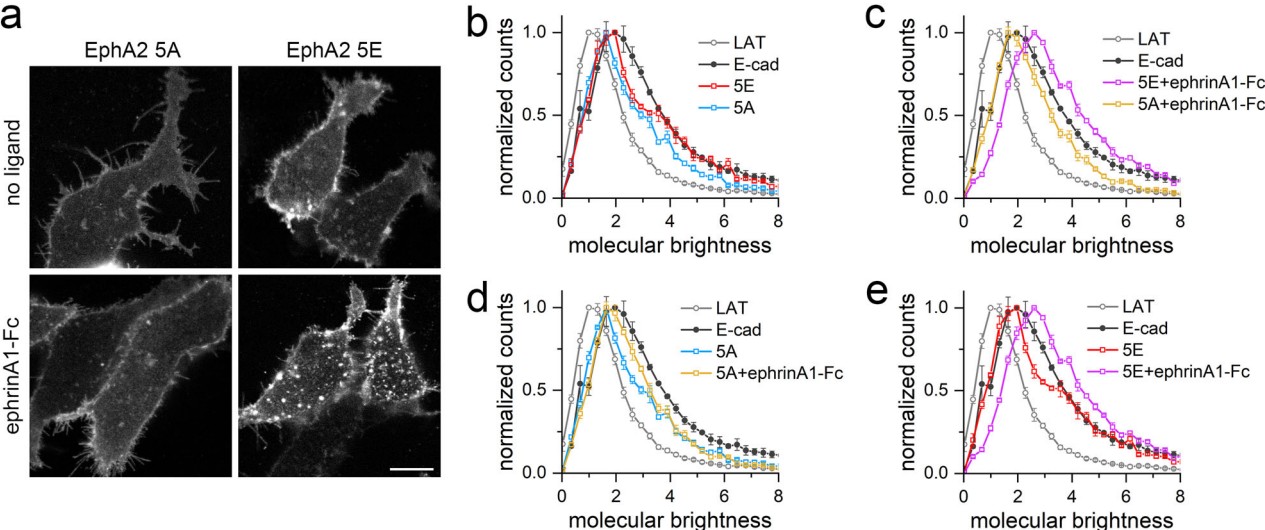

**Fig. 6 FIF analysis of oligomerization for EphA2 kinase-SAM linker mutants. a** Representative images of HEK293T cells transiently transfected with EphA2 5A-EYFP or EphA2 5E-EYFP and stimulated with ephrinA1-Fc or unstimulated (no ligand). More fluorescent patches appear to be present in the cells expressing the EphA2 5E mutant and stimulated with ephrinA1-Fc compared to the other conditions. Scale bar = 10 μm. **b** Normalized FIF brightness distributions for the EphA2 5A ($n = 134$) and 5E ($n = 129$) mutants in the absence of ligand treatment. Data are compared to previously published monomer (LAT) and dimer (E-cad, E-cadherin) controls[47]. The brightness distribution curves for EphA2 5A and 5E are between the two controls, suggesting that both mutants exist in a monomer-dimer equilibrium. **c** Normalized FIF distributions for the EphA2 5A ($n = 122$) and 5E ($n = 128$) mutants stimulated with a saturating concentration of ephrinA1-Fc. The brightness distribution curve for the EphA2 5A mutant is between the two control curves, whereas the EphA2 5E mutant distribution curve is shifted to larger brightness values than the E-cadherin curve, suggesting the formation of higher-order oligomers. **d, e** Comparison of the normalized FIF distributions for **d** the EphA2 5A mutant and **e** the EphA2 5E mutant with and without ephrinA1-Fc. EphrinA1-Fc causes only a small shift in the curves for the EphA2 5A mutant but a large shift in the curve for the EphA2 5E mutant, suggesting that the EphA2 5E mutant can much more readily form higher order oligomers. The counts of molecular brightness were normalized, averaged, and plotted along with their standard errors in **b–e**. Source data for **b**, **c**, **d**, **e** are provided in the Source Data file.

phosphorylation of S897, which creates the necessary negatively charged motif[27]. This was confirmed in in vitro kinase assays with several CK1 family members, which did not efficiently phosphorylate peptide substrates in which the phosphorylated S897 was replaced by alanine (Fig. 7c). In addition, phosphorylation of peptides with an S892A substitution revealed that most kinases identified in the screen preferentially phosphorylate S892 (Fig. 7d). We have previously shown that PKC family members play an important role in S892 phosphorylation in cells[28].

To identify kinases that phosphorylate T898 and/or S899, we used an ADP-Glo assay with two peptide substrates in which these two residues are the only possible phosphorylation sites (Fig. 7e). One of the peptides contains phosphorylated S892, S897 and S901, to identify kinases (such as acidophilic kinases) that can generate a fourth and fifth phosphosite in the cluster. In the other peptide the three Ser residues are replaced by alanine, to identify kinases that can phosphorylate T898 and/or S899 in the absence of other phosphorylated residues. Among the acidophilic kinases (Fig. 7e), we found that CK2 family kinases preferentially phosphorylate the peptide with three other phosphosites, suggesting that once S892, S897, and S901 are phosphorylated, the constitutively active CK2 can further increase linker phosphorylation. CK2 kinases optimally recognize substrates with a phosphosite or a negative charge at the +3 position, such as T898 when S901 is phosphorylated and S899, which is upstream of the negatively charged E902[48,49]. CK2 kinases were not among the most active kinases in the initial screen with the S897 phosphorylated peptide (Supplementary Table 3). However, S897 phosphorylation may not be sufficient to promote phosphorylation of other cluster residues by CK2 (Fig. 7e). Some CK1 family members can also target T898 and/or S899, while the GSK3 and PLK families show low activity (Fig. 7e).

Other (non-acidophilic) kinases were chosen for testing (Fig. 7e) because of their residual activity towards the peptide containing T898, S899 and S901 as the only phosphorylatable residues (Fig. 7d). These kinases include NEK2, NEK7, ULK1, ULK2, MAP3K9, members of the PKC family, and CAMK2B. Additional kinases were chosen based on the prediction scores from the PhosphoNET Kinase Predictor online resource (phosphonet.ca), which considers only non-phosphorylated sequence motifs. The T898 motif has very low prediction scores, suggesting that this residue is unlikely to be phosphorylated when none of the other linker residues is phosphorylated. The top predicted kinases for the S899 motif are those of the PIM, PKA, p70S6K, AKT, SGK, RSK and PKG families. Some of these kinases substantially phosphorylated T898 and/or S899, with several of them preferring the peptide substrate lacking other phosphorylation sites (Fig. 7e). These data suggest that T898, and possibly S899, can be phosphorylated by acidophilic kinases once other residues in the cluster are phosphorylated. In addition, S899 and possibly T898 may also be phosphorylated by other kinases that do not require surrounding negative charges. Conceivably, once bound to EphA2 to phosphorylate S892 or S897, kinases such as PKC, RSK, AKT and PKA may processively phosphorylate additional residues[50–52].

We also used a peptide phosphorylatable only on S897 to test kinases predicted by PhosphoNET to phosphorylate the S897 motif. This revealed new candidate S897 kinases, and most notably members of the CAMK2 family (Fig. 7f). Some of the kinases that phosphorylate the peptide containing only S897, show better activity towards a similar peptide that also contains the other four serine/threonine residues and/or still phosphorylate a peptide in which S897 is already phosphorylated. Thus, these kinases may also be able to phosphorylate other linker residues.

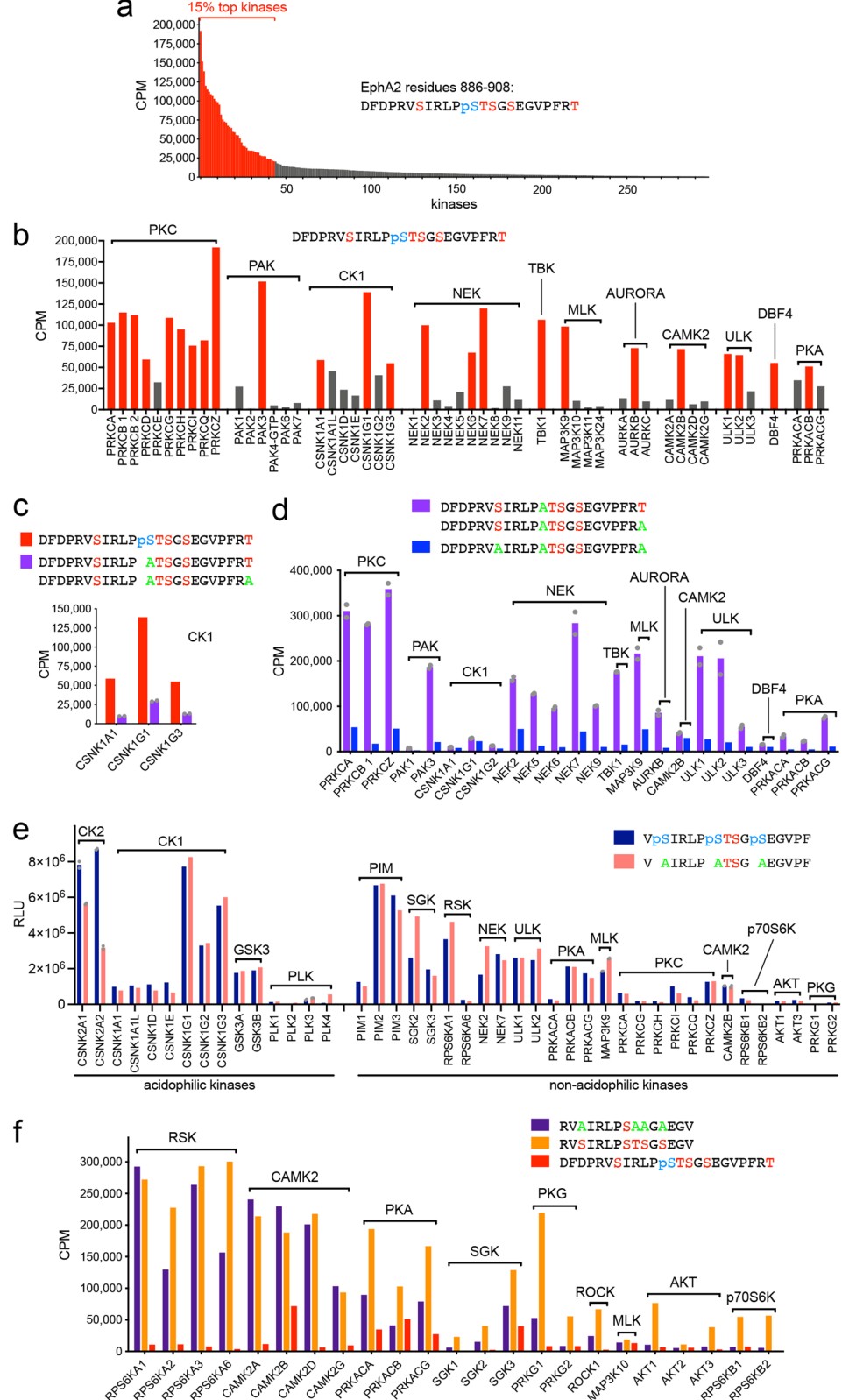

The model emerging from our data is that several kinases can phosphorylate the five serine/threonine residues in the EphA2 kinase-SAM linker, leading to concomitant phosphorylation of multiple residues that likely coordinately affect EphA2 signaling function by regulating the conformation of the EphA2 intracellular region.

## Discussion

How the different parts of the Eph receptor intracellular region— including the juxtamembrane segment, kinase domain, linker and SAM domain—are arranged and function together has remained an open question, in part due to lack of structural information that extends beyond single domains and adjacent regions[14,37,53].

**Fig. 7 Multiple kinases phosphorylate the five serine/threonine residues in the EphA2 kinase-SAM linker. a** Screen of 298 kinases using in vitro kinase reactions with [γ-$^{33}$P] ATP and a peptide substrate containing S892, T898, S899, S901 and phosphorylated S897 identifies many kinases that phosphorylate residues other than S897. The 44 kinases (15% of those screened) mediating the highest $^{33}$P incorporation are shown in red. **b** The 24 kinases mediating the highest $^{33}$P incorporation (>50,000 CPM) are shown in red and other less active family members that were also tested are shown in gray. Kinase families are ordered based on the most active family member. **c** CK1 family members preferentially phosphorylate the peptide with phosphorylated S897 compared to two peptides with Ala replacing S897 and differing in the last residue (averaged together). **d** Comparison of two peptides containing S892 and differing in the last residue (averaged together) with a peptide in which S892 is replaced by Ala shows that many of the kinases identified in the screen mainly phosphorylate S892. Kinase families are ordered as in **b**. **e** Comparison of the phosphorylation of the two indicated peptides identifies kinases that can phosphorylate T898 and/or S899 in the presence or absence of prior S892, S897 and S901 phosphorylation. **f** $^{33}$P incorporation into peptides in which only S897 can be phosphorylated identifies kinase families that can phosphorylate this residue. Higher $^{33}$P incorporation into the peptide also containing the other four serine/threonine residues and residual $^{33}$P incorporation into the peptide with already phosphorylated S897 suggest that some of the kinases can also phosphorylate other residues. Individual data points in c and d (for the measurements with two peptides differing only in the last residue) and in e (for duplicate measurements with CK2 kinases, PLK3, MAP3K9 and CAMK2B) are shown as gray dots. CPM, counts per minute measuring incorporated $^{33}$P; RLU, relative light units. Graphs were generated using Prism software (GraphPad). Source data for the in vitro kinase reactions are provided in Supplementary Table 3.

A key question is how phosphorylation of the EphA2 linker regulates non-canonical signaling in the cell. Linkers are known to play a crucial role in the propagation of information between protein modules by communicating conformational and/or dynamic alterations[54–57]. Due to their flexible nature, they can serve as hinges, enabling reorientation of protein domains with respect to each other to regulate protein function. Length, amino acid composition and post-translational modifications, such as phosphorylation, can profoundly impact the conformation and dynamics of linkers, affecting their behavior.

Our integrative structural biology characterization of the EphA2 intracellular region provides new insights into its conformation, dynamics and regulation. Our data, based on introduction of glutamic acid as a proxy for serine/threonine phosphorylation, suggest a model for the conformational dynamics that underlie EphA2 non-canonical signaling induced by cumulative phosphorylation of the kinase-SAM linker. It should be noted that glutamic acid is associated with a lower negative charge than a phosphate group and that five glutamic acid residues approximately mimic three phosphate groups in terms of the negative charge introduced[58]. Our HDX data in combination with interactions observed in our crystal structures suggest that in the absence of linker phosphorylation the intracellular portion of EphA2 favors more closed conformations that at least in part depend on interactions between the electronegative surface of the kinase domain and the electropositive surface of the SAM domain (Fig. 8). Cumulative effects of the negative charges mimicking linker phosphorylation shift the equilibrium towards more open conformations with decreased interaction between kinase and SAM domains. The model that the linker is flexible, and its phosphorylation regulates the conformation of the EphA2 intracellular region, is supported by the variation in the relative arrangement of the two domains observed in our crystal structures and SAXS experiments. Additionally, we found that the E820K-E825K double mutation, which disturbs the electronegative surface of the EphA2 kinase domain, and thus likely its interaction with the SAM domain, has similar effects on HDX as multiple phosphomimetic mutations in the kinase-SAM linker. We speculate that the cumulative effects of multiple phosphorylation sites in the linker represent a barcode that allows fine-tuning of non-canonical signaling strength by progressively shifting the EphA2 conformational equilibrium towards more open conformations.

Our data also suggest a potential interplay between non-canonical and canonical signaling through E820 in the αFG loop of the kinase domain, which may function as a sensor for S892 phosphorylation and, to a lesser extent, S897 phosphorylation. Electrostatic repulsion between E820 and negatively-charged/

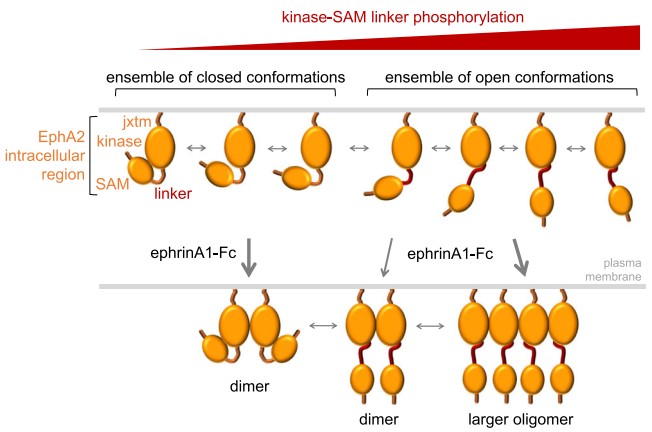

**Fig. 8 Hypothetical model illustrating the proposed effects of phosphorylation on the conformation of the EphA2 intracellular region.** In the unphosphorylated, inactive EphA2 intracellular region, the kinase domain interacts with the SAM domain (SAM). Phosphorylation of serine/ threonine residues in the kinase-SAM linker promotes transition to more open conformations. The ephrinA1-Fc ligand induces EphA2 dimerization when phosphorylation of the kinase-SAM linker is low and EphA2 clustering into larger oligomers when linker phosphorylation is high.

phosphorylated residues at positions 892 or 897 could at least in part explain why in HDX experiments the region around E820 becomes more solvent exposed in the EphA2 S892E mutant and to a lesser extent in the S897E mutant. Supporting this hypothesis, E820 is in a different conformation in the EphA2 WT structure compared to the EphA2 S897E/S901E structure (Supplementary Fig. 9). Further support comes from previous studies on EphA3. In an EphA3 structure lacking the SAM domain[37], the equivalent E827 residue points away from the linker (Supplementary Fig. 9), which includes the negatively-charged D904 at the position corresponding to S897[37]. An allosteric linkage of the EphA3 kinase-SAM linker with the αFG loop, activation loop and juxtamembrane segment mediated by E827 and W826 (corresponding to EphA2 E820 and W819) has been previously proposed[14]. Furthermore, in vitro kinase reactions with the recombinant EphA3 intracellular region lacking the SAM domain suggest that the same allosteric network is involved in the regulation of receptor autophosphorylation on tyrosine residues[14], implying that canonical and non-canonical signaling may influence each other. Similarly, in EphA2 changes in E820 and the αFG loop may be relayed to the activation loop (located just above E820; Supplementary Fig. 9), the juxtamembrane segment, and the SAM domain (which may interact with the negatively-

charged surface formed by the αFG loop and adjacent regions). These observations suggest that the E820K mutation has multiple effects on the configuration of the E820K/E825K mutant intracellular region, not only weakening binding of the SAM domain to the kinase domain but also disrupting the long-range regulatory network involving the kinase-SAM linker.

In principle, there could be other interpretations of our data. For example, the time-dependent increase in HDX observed in regions of the kinase and SAM domains (Fig. 4a) may be due to the biophysical properties of these regions, both of which contain expansive highly charged surfaces, rather than to their reciprocal interaction shielding them from the solvent. Furthermore, the observed changes in HDX induced by negative charges in the linker (Fig. 5) could reflect independent conformational changes within the kinase and SAM domains, rather than an interaction between the two domains, or changes in hydrogen-bonding propensities rather than changes in solvent accessibility[40,41]. We believe these alternative possibilities are unlikely for several reasons. First, HDX occurring on a time scale of a few minutes, preferentially reports on the solvent accessibility of amide protons not engaged in stable hydrogen bonding[41]. Second, we observed correlated HDX rates in two regions located in different domains connected by a flexible linker. The two regions show similar HDX in EphA2 WT (Fig. 4a) and in EphA2 linker mutants (Fig. 5). It seems very unlikely that these correlated HDX changes are due to independent changes in hydrogen bonding or in the conformations of the two domains, especially since the mutations introduced are in the flexible linker connecting the domains rather than within the domains. Finally, the two regions have opposite charges, consistent with their potential interaction. In fact, we observed an interaction between the two regions from molecules A and B in the crystal structure of EphA2 WT. This intermolecular interaction was observed at the high protein concentrations in the crystals, whereas in the HDX experiments the lower concentration of the EphA2 intracellular region in solution would favor an intramolecular interaction. This intramolecular interaction would be expected to be physiologically relevant in the cell, where EphA2 concentrations are also much lower than in the crystals.

EphA2 non-canonical signaling involving S897 phosphorylation was discovered more than ten years ago[22]. Although S897 is part of a cluster of five phosphorylated residues, prior studies have focused essentially only on this serine[26]. The importance of S897 phosphorylation is highlighted by the effects of the S897A mutation, which drastically impairs EphA2 non-canonical signaling[22,27,33]. Whether S897 phosphorylation might have effects through mechanisms other than contributing to cumulative negative charges in the linker remains to be determined, given the disagreement on the functional effects of the mutation of only S897 to a negatively charged residue[22,59,60]. Our findings suggest that the role of S897 phosphorylation in regulating EphA2 conformation does not depend only on the modification of this single residue but also involves promoting phosphorylation of other linker residues. Phosphosites in a cluster can accumulate until a response threshold is reached, enable gradual effects, and/ or potentiate the robustness of phosphorylation-dependent responses. Bulk electrostatic effects seem to be an important mechanism through which clusters of phosphorylated residues in unstructured regions, such as linkers, can affect protein function, particularly when the number of the phosphosites appears to be more critical than their exact positions[51,52,61]. This seems to be the case for the EphA2 kinase-SAM linker, where many different combinations of two or three of the five possible phosphorylation sites are observed. We hypothesize that EphA2 non-canonical signaling is regulated by concomitant phosphorylation of multiple linker residues. Four of the five phosphorylation sites are

conserved in another Eph receptor, EphA1, supporting the functional importance of the phosphorylation cluster and the ability of EphA1 to also mediate non-canonical signaling.

Several mechanisms could lead to the observed concomitant phosphorylation of multiple residues in the EphA2 linker. First, phosphorylation of some linker residues can promote phosphorylation of additional residues by acidophilic kinases[48,51,62]. A second mechanism likely involves processive phosphorylation[50–52]. Perhaps consistent with this, we have previously found that PKA activation increases EphA2 linker phosphorylation on multiple residues[27] and we show here that PKA and other kinases are able to phosphorylate multiple linker residues (Fig. 7). A third mechanism is the coincident phosphorylation of different residues by kinases that are concomitantly regulated, for example by oncogenic transformation or because they are part of the same signaling network. Together, these findings imply that regulation of EphA2 linker phosphorylation leading to non-canonical signaling is more complex than previously envisioned. Accordingly, we have identified multiple kinases that can phosphorylate each serine/threonine residue in the EphA2 linker in in vitro kinase reactions (Supplementary Table 3). While a number of these kinases have already been shown to phosphorylate EphA2 in cells[22,26–28,33], additional studies are needed to further define the repertoire of kinases that phosphorylate EphA2 to regulate non-canonical signaling in response to different stimuli.

Our FRET and FIF data reveal differences in the oligomerization behavior of EphA2 mutants differing in linker negative charges, suggesting that linker phosphorylation plays a role in regulating EphA2 lateral interactions on the cell surface. While the effects of linker negative charges on EphA2 oligomerization in the absence of ligand are small, the effects in the presence of ligand are pronounced. Our FIF experiments show that the EphA2 5E mutant associates into larger oligomers in the presence of ephrinA1-Fc, whereas the EphA2 5A mutant remains dimeric. Thus, intramolecular interaction of the SAM domain with the kinase domain in closed configurations may impair EphA2 lateral association, perhaps by preventing intermolecular interactions between the positively and negatively charged surfaces on opposite sides of the EphA2 kinase domain. This observation suggests another potential link between non-canonical and canonical signaling, although further work is needed to study potential structural effects of multiple alanine mutations in the EphA2 linker. The EphA2 open conformations promoted by linker phosphorylation could also expose kinase and SAM domain surfaces not readily accessible in the closed conformations for the binding of effector proteins that mediate non-canonical signaling[25,60,63].

It will be important in future experiments to further investigate the different forms of interplay between EphA2 canonical and non-canonical signaling. Numerous studies show that in certain cellular contexts EphA2 canonical signaling can downregulate S897 phosphorylation through inhibition of ERK-RSK and AKT[20,22,23,25,27,64], while a recent report suggests that EphA2 kinase-SAM linker phosphorylation can inhibit canonical signaling in embryonic stem cells[65]. Conversely, canonical signaling does not inhibit EphA2 S897 phosphorylation mediated by PKA or in cells harboring activating mutations in the ERK MAP kinase pathway[27,33]. It appears that depending on the cellular context, EphA2 canonical and non-canonical signaling can be mutually exclusive or occur simultaneously and perhaps modulate each other.

In conclusion, we have shown that the EphA2 kinase-SAM linker can be simultaneously phosphorylated on multiple serine/ threonine residues and identified kinases able to phosphorylate these residues. Furthermore, we have uncovered how cumulative phosphorylation on multiple linker residues can affect the conformation of the EphA2 intracellular region. Since EphA2

linker phosphorylation has been widely implicated in cancer malignancy, our findings can inform further mechanistic studies of EphA2 function and therapeutic strategies to target EphA2 signaling. Our findings also have implications for the functional regulation of EphA1 and for the structural understanding of the intracellular region of other Eph receptors.

## Methods

**Antibodies**. Antibodies recognizing EphA2 (rabbit mAb #6997) and the phosphorylated S897 motif (rabbit mAb #6347) were purchased from Cell Signaling Technology and used at a dilution of 1:1,000 for immunoblotting. Affinity purified rabbit antibodies recognizing the phosphorylated S892 or S901 motifs were generated and used at 1 μg/ml for immunoblotting. The EphA2 antibody was confirmed to label transiently transfected EphA2 and not other Eph receptors. The phosphospecific antibodies were validated by showing that they recognize EphA2 WT by immunoblotting, with a stronger signal under conditions promoting kinase-SAM linker phosphorylation, but do not recognize EphA2 in which the Ser in the relevant phosphorylated motif is mutated to Ala (S892A, S897A, or S901A)[27,28]. In addition, the phosphospecific antibodies were shown to label transiently transfected EphA2 WT but not to recognize the corresponding phosphorylated motif in the EphA1 kinase-SAM linker. Goat-anti-rabbit secondary antibodies conjugated to HRP (A16110; Invitrogen/ThermoFisher Scientific) were used at a dilution of 1:3,000.

**Cell lines**. The following cell lines were purchased from the American Type Culture Collection (Manassas, VA): PC3 androgen-independent prostate cancer (CRL-1435), A549, Calu-3 and H1648 lung adenocarcinoma (CCL-185, HTB-55 and CRL-5882 respectively), HCC1937 triple negative breast cancer (CRL-2336), SW626 colon adenocarcinoma (HTB-78), SKOV3 ovarian serous cystadenocarcinoma (HTB-77) and HEK293T human embryonic kidney (CRL-3216). The Mel-Juso melanoma cell line was purchased from the DSMZ (ACC 74); the HOP62 lung adenocarcinoma cell line from the tumor/cell line repository of the Developmental Therapeutics Program, Division of Cancer Treatment and Diagnosis, National Cancer Institute. The BxPC3 pancreatic cancer cell line was kindly provided by P. Itkin-Ansari (Sanford Burnham Prebys Medical Discovery Institute), the Panc1 cell line by F. Levine (Sanford Burnham Prebys Medical Discovery Institute); the U87, T98G and U251-MG glioblastoma cell lines by W. Stallcup (Sanford Burnham Prebys Medical Discovery Institute); the BT549 breast cancer cell line by R. Maki (Sanford Burnham Prebys Medical Discovery Institute); the MDA-MB-468 breast cancer cell line by K. Vuori (Sanford Burnham Prebys Medical Discovery Institute); and the MDA-MB-231 breast cancer cell line by J. Smith (Sanford Burnham Prebys Medical Discovery Institute). The Mel-Juso cell line was stably infected with the pLVX-IRES-Neo lentivirus encoding EphA2 wild-type with an N-terminal FLAG tag[27].

The PC3, MDA-MB-231, BT549, T98G, U87 and U251-MG cell lines were authenticated after completion of the experiments and Panc-1 several passages before use in the experiments by performing short tandem repeat analysis on isolated genomic DNA with the GenePrint® 10 System (Promega), and peaks were analyzed using GeneMarker HID (Softgenetics). Allele calls were searched against short tandem repeat databases[28]. Other cell lines were used after few passages to expand cells obtained from the vendor and thus not authenticated. BxPC3 and MDA-MB-468 were not authenticated.

A549, BxPC3, H1648, HCC1937, HOP62, PC3 and Mel-Juso cells were cultured in RPMI 1640 medium (Gibco); BT549, Calu-3, T98G, MDA-MB-231, MDA-MB-468, Panc1, SW626, U251-MG and U87 cells were cultured in Dulbecco's Modified Eagle Medium (DMEM; Corning) and SKOV3 cells were cultured in McCoy's containing 10% FBS. All culture media contained 10% fetal bovine serum, antimycotics and antibiotics. HEK293T cells used in FRET experiments were cultured in DMEM supplemented with 10% fetal bovine serum, 3.5 g l$^{-1}$ D-glucose, and 1.5 g l$^{-1}$ sodium bicarbonate.

**Kinase reactions**. The peptide substrates were synthesized by Kinexus (Vancouver, Canada) at >95% or > 98% purity. The recombinant protein kinases were cloned, expressed and purified using Kinexus proprietary methods and underwent routine quality control testing. The assay conditions were optimized to yield acceptable enzymatic activity. In addition, the assays were optimized to give high signal-to-noise ratio.

Protein kinase assays were performed by Kinexus, in most cases using a radioisotope assay format. These assays were performed in a final volume of 25 μl, including 5 μl diluted active kinase (~10-50 nM final concentration), 5 μl stock peptide substrate solution (500 μM final concentration), 10 μl kinase assay buffer, 5 μl [γ-$^{33}$P]ATP (250 μM stock solution, 0.8 μCi). The assay was initiated by the addition of [γ-$^{33}$P]ATP and the reaction mixture incubated at room temperature for 30 min. The assay was terminated by spotting 10 μl of the reaction mixture onto a multiscreen phosphocellulose P81 plate, which was then washed 3 ×15 min in a 1% phosphoric acid solution. The radioactivity on the P81 plate was measured in the presence of scintillation fluid in a Trilux scintillation counter.

Some protein kinase assays were performed using the ADP-Glo™ assay kit (Promega), which measures the generation of ADP in a kinase reaction. Generation of ADP leads to an increase in luminescence signal in the presence of ADP-Glo™. The protein kinase assays were performed at 30 °C for 30 min in a final volume of 25 μl, including 5 μl diluted active protein kinase to obtain a previously determined optimal concentration, 5 μl of a 125 μM stock peptide substrate solution (25 μM final concentration), 5 μl kinase assay buffer, 5 μl 10% DMSO, 5 μl ATP stock solution. The assay was started by incubating the reaction mixture in a 96-well plate at room temperature for 30 min. The assay was terminated by the addition of 25 μl of ADP-Glo™ Reagent (Promega). The 96-well plate was shaken and then incubated for 40 min at room temperature. Then, 50 μl of Kinase Detection Reagent was added, the 96-well plate was shaken and then incubated for further 30 min at room temperature. The 96-well reaction plate was read using the ADP-Glo™ Luminescence Protocol on a GloMax plate reader (Promega; Cat# E7031).

**Immunoprecipitation and immunoblotting**. Cells were collected when they reached ~90% confluency. For immunoprecipitation, cells were collected in ice-cold modified RIPA buffer (50 mM Tris-HCl pH7.6, 150 mM NaCl, 1% Triton X-100, 0.5% sodium deoxycholate, 0.1% SDS, and 2 mM EDTA) supplemented with Halt Protease and Phosphatase Inhibitor cocktail (PI78442; Thermo Fisher Scientific). Lysates were centrifuged at 13,000 g for 10 min at 4 °C to remove insoluble material. Supernatants were pre-cleared by incubation with Sepharose beads (4B200; Sigma Aldrich) for 15 min at 4 °C on a rotator. Each immunoprecipitation was performed using 6 μg EphA2 antibody (05-480; EMD Millipore) coupled to 20 μl GammaBind Plus sepharose beads (17-0886-02; GE Healthcare) for 2 h at 4 °C. Immunoprecipitates were washed three times with ice-cold modified RIPA buffer and once with phosphate buffered saline (PBS), and eluted by heating at 95 °C for 2 min in SDS-PAGE sample buffer. For immunoblotting, cells were collected in SDS-PAGE sample buffer and lysates were heated at 95 °C for 2 min followed by a brief sonication. After SDS-PAGE and transfer to a PVDF membrane, blots were incubated for 1 h in blocking buffer (0.1% Tween-20, 5% BSA in PBS) and then overnight at 4 °C with primary antibodies diluted in blocking buffer. Membranes were then incubated for 1 h at room temperature with goat-anti-rabbit secondary antibodies conjugated to HRP followed by ECL chemiluminescence detection (RNP2106; GE Healthcare Bio-Sciences, Piscataway, NJ) using a ChemiDoc gel imaging system (Bio-Rad), and analyzed using Prism software (GraphPad). Uncropped blots are provided in the Source Data file.

**Recombinant EphA2 expression and purification**. The cDNA sequence coding for the intracellular portion of human EphA2 (residues 590–976) was cloned into the pETNKI-LIC vector, which encodes an N-terminal 6xHis-tag followed by a 3C protease cleavage site in a pET29 vector backbone. Mutations were introduced in constructs using standard site-directed mutagenesis. The sequences of all PCR-amplified portions of constructs were verified by Sanger sequencing (Retrogen, San Diego).

EphA2 was expressed in *E. coli* BL21(DE3) grown in 2xYT medium (BD Difco) at 20 °C overnight and purified using Ni-NTA agarose (Qiagen) followed by cleavage of the His-tag with 3C protease and dephosphorylation with bovine alkaline phosphatase (Sigma). Further purification was performed by ion exchange chromatography (Source Q for WT and the S897E, S897E/S901E, and E820K/E825K mutants and Source S for the 3E and 5E mutants) equilibrated in 10 mM NaCl, 10 mM HEPES pH 7.9 and eluted with a linear NaCl gradient up to 500 mM. EphA2 was concentrated to 5–7 mg ml$^{-1}$. For HDX-MS, the cDNA sequence coding for the intracellular portion of human EphA2 (residues S570-D976) was cloned into the pET29a(+) vector (Novagen) to express EphA2 protein with a C-terminal 6xHis-tag. The protein was purified as described above, but no 3C protease cleavage was performed and the protein was further purified on a Superdex 200 Increase 10/300 GL size-exclusion chromatography column (GE Healthcare) equilibrated in 100 mM NaCl, 10 mM HEPES pH 7.9. All proteins were flash frozen in ~100 μl aliquots in liquid nitrogen and stored at −80 °C.

**Crystallization and structure solution**. For crystallization, EphA2 WT or S901E (residues D590-I976, ~7 mg ml$^{-1}$) were mixed with 1.7 mM β,γ-methyleneadenosine 5′-triphosphate (AMPPCP) and 10 mM MgCl$_2$, and initial crystals were obtained with the Hampton Index HT screen at 14 °C. Crystals were optimized by increasing drop size and varying protein to precipitant ratio. Crystals of the S897E/S901E mutant (~5 mg ml$^{-1}$) were additionally optimized using the Hampton Additive Screen HT. Final crystallization conditions are 0.2 M MgCl$_2$, 0.1 M Bis-Tris pH5.5, 25% PEG 3,350 for EphA2 WT and EphA2 S901E and 4.5% tacsimate, 0.09 M HEPES pH 7.0, 9% PEG-MME 5 K, 0.1 M CsCl for EphA2 S897E/S901E (Supplementary Table 1). Crystals were cryoprotected by step-wise transfer to reservoir solutions supplemented with 5-15% glycerol and cryocooled in a nitrogen stream at 100 K. Diffraction data were collected using Rigaku CrystalClear 2.0 on a rotating anode X-ray generator (Rigaku FR-E superbright, 1.54 Å wavelength) at 100 K and processed in XDS[66] or MOSFLM[67] and with software from the CCP4 suite[68]. Phases were obtained using molecular replacement in Phaser[69] with the EphA2 kinase domain from PDB 4PDO and the EphA2 SAM domain from PDB 3KKA as search models. Model building and refinement were respectively performed in Coot[70] and Phenix[71]. Crystallographic models and data have been

deposited in the PDB under accession numbers 7KJA (EphA2 WT residues 590–976 [https://doi.org/10.2210/pdb7KJA/pdb]), 7KJB (EphA2 S897E/S901E residues 590–976 [https://doi.org/10.2210/pdb7KJB/pdb]) and 7KJC (EphA2 S901E residues 590–976 [https://doi.org/10.2210/pdb7KJC/pdb]). Data collection and refinement statistics are reported in Supplementary Table 1. Ramachandran statistics are as follows: EphA2 WT: 98.9% favored, 1.1% allowed; EphA2 S897E/S901E: 97.2% favored, 2.8% allowed; EphA2 S901E: 97.2% favored, 2.8% allowed. Figures were prepared in PyMOL 2.3.4 (Schrödinger).

**Size exclusion chromatography–small angle X-ray scattering (SEC-SAXS).** EphA2 proteins were dialyzed into 150 mM NaCl, 10 mM HEPES pH 7.9 and their concentration was adjusted to 5 mg mL$^{-1}$. SAXS data were collected at the Argonne National Laboratory Advanced Photon Source Biocat SAXS/WAXS beamline 18-ID, using an inline size-exclusion chromatography set-up (AKTA pure with Superdex 200 10/300 GL column; Cytiva) and at the Australian synchrotron SAXS/WAXS beamline, using an inline size-exclusion chromatography set-up (Agilent FPLC with Superdex 200 increase 5/150 GL column; Cytiva) in 10 mM HEPES pH 7.9, 150 mM NaCl, 5% glycerol. Two-dimensional scattering data were radially averaged, normalized to sample transmission, and scatter patterns from the apex of the size-exclusion chromatography elution peak were averaged. Background scatter (scatter patterns from early in the SEC run) was subtracted using the Scatterbrain software package (Australian Synchrotron). The ATSAS suite of software[72] was used to perform Guinier analysis (PRIMUS) to calculate the pairwise distance distribution function $P(r)$ and the maximum particle dimension $D_{max}$ (GNOM) and to generate the bead models (DAMMIF, DAMAVER, and DAMMIN). Bead models and crystal structures were superimposed using the ATSAS subcomb script. The molecular mass of each sample was estimated using the SAXS-MoW2 package[73]. All structural models were illustrated using PyMOL 2.3.4 (Schrödinger). All data collection and processing statistics are summarized in Supplementary Table 2.

**Hydrogen-deuterium exchange–mass spectrometry (HDX-MS).** HDX-MS was performed at the Biomolecular and Proteomics Mass Spectrometry Facility (BPMSF) of the University California San Diego, using a Synapt G2Si system with HDX technology (Waters Corporation) according to methods previously described[74]. Briefly, deuterium exchange reactions were performed using a Leap HDX PAL autosampler (Leap Technologies, Carrboro, NC). D$_2$O buffer was prepared by lyophilizing 10 mM HEPES pH 7.9, 100 mM NaCl initially dissolved in ultrapure water (H$_2$O buffer) and redissolving the powder in the same volume of 99.96% D$_2$O (Cambridge Isotope Laboratories, Inc., Andover, MA) immediately before use. Deuterium exchange was measured in triplicate at each time point (0 min, 0.5 min, 1 min, 2 min, 5 min). For each deuteration time point, 4 µl of protein (5 µM in H$_2$O buffer) were held at 25 °C for 5 min before being mixed with 56 µl of D$_2$O buffer. The deuterium exchange was quenched for 1 min at 1 °C by combining 50 µl of the deuteration reaction with 50 µl of 250 mM tris(2-carboxyethyl)phosphine (TCEP) pH 2.5. The quenched sample (90 µl) was then injected in a 100 µl sample loop, followed by digestion on an in-line pepsin column (Immobilized Pepsin, Pierce) at 15 °C. The resulting peptides were captured on a BEH C18 Vanguard precolumn, separated by analytical chromatography (Acquity UPLC BEH C18, 1.7 µm 1.0 × 50 mm, Waters Corporation) using a 7–85% acetonitrile gradient in 0.1% formic acid over 7.5 min, and electrosprayed into the Waters Synapt G2Si quadrupole time-of-flight mass spectrometer. The mass spectrometer was set to collect data in the Mobility, ESI + mode; mass acquisition range of 200–2000 (m/z); scan time 0.4 s. Continuous lock mass correction was accomplished with infusion of leu-enkephalin (m/z = 556.277) every 30 s (mass accuracy of 1 ppm for calibration standard).

For peptide identification, the mass spectrometer was set to collect data in mobility-enhanced data-independent acquisition (MS$^E$), mobility ESI + mode. Peptide masses were identified from triplicate analyses and data were analyzed using the ProteinLynx global server (PLGS) version 2.5 (Waters Corporation). Peptide masses were identified using a minimum number of 250 ion counts for low energy peptides and 50 ion counts for their fragment ions; the peptides also had to be larger than 1,500 Da. The following cutoffs were used to filter peptide sequence matches: minimum products per amino acid of 0.2, minimum score of 7, maximum MH + error of 5 ppm, and a retention time RSD of 5%. In addition, the peptides had to be present in two of the three ID runs collected. The peptides identified in PLGS were then analyzed using DynamX 3.0 data analysis software (Waters Corporation). Peptides containing mutated residues were manually assigned. The relative deuterium uptake for each peptide was calculated by comparing the centroids of the mass envelopes of the deuterated samples with the undeuterated controls following previously published methods[75]. The peptides reported on the coverage maps are those from which data were obtained. For all HDX-MS data, 3 technical replicates were analyzed. The deuterium uptake was corrected for back-exchange using a global back exchange correction factor (typically ~25%) determined from the average percent exchange measured in disordered termini of various proteins[76]. Deuterium uptake plots in Supplementary Fig. 5 and coverage maps in Supplementary Fig. 6 were generated in DECA (github.com/komiveslab/DECA) and the data in Supplementary Fig. 5 are fitted with an exponential curve for ease of viewing[77]. Relative fractional deuterium uptake plots were plotted using Prism (GraphPad).

**Analytical ultracentrifugation.** Sedimentation velocity experiments were performed using a ProteomeLab XL-I (Beckman Coulter) analytical ultracentrifuge. EphA2 590-976 sample with OD$_{280nm}$ = 0.5 (~9 µM) in 10 mM HEPES pH 7.9, 100 mM NaCl was loaded into 2-channel cells and spun in an An-50 Ti 8-place rotor at 40,000 rpm (~116,500 g at the center of the cell) and 20 °C for 15 h. Data were analyzed using Sedfit software (P. Schuck, NIH/NIBIB)[78].

**Size-exclusion chromatography coupled with multi-angle light scattering (SEC-MALS).** SEC-MALS experiments were performed using a Superdex 200 increase 10/300 SEC column (Cytiva), equilibrated in 20 mM HEPES pH 7.0, 150 mM NaCl connected to an HPLC system (Shimadzu Corporation) and DAWN HELEOS-II (Wyatt Technology) light scattering detector and Optilab T-rEX (Wyatt Technology) refractive index detector. 100 µl EphA2 570-976 at a concentration of 3 mg/ml was injected. Data were analyzed in ASTRA software (Version 7.3.1.9, Wyatt Technology).

**FRET imaging and analysis.** HEK293T cells were seeded at a density of 2×10$^5$ cells/dish in 35 mm glass bottom collagen-coated dishes (MatTek Corporation) and cultured for 24 h at 37 °C in 5% CO$_2$. Dimerization experiments were performed in cells transiently transfected with EphA2 labeled at the C-terminus with fluorescent proteins (either mTurquoise or EYFP, a FRET pair) attached via a flexible linker (GGS)$_5$. The EphA2 5E, 3E, S897E, S892E, 5A (S892A/S897A/T898A/S899A/ S901A), ASAAA (S892A/T898A/S899A/S901A), S897A, or S892A mutants in pcDNA3 were transfected using the Lipofectamine 3000 reagent (ThermoFisher Scientific/Invitrogen). Cells were transfected with 0.5-2 µg total DNA at a 1:2 EphA2-mTurquoise:EphA2-EYFP ratio. Twelve hours later, the cells were washed twice with starvation medium to remove traces of phenol red, and serum-starved overnight to ensure no soluble ligands were present.

FRET experiments were performed following published protocols[43]. HEK293T cells expressing full-length EphA2 WT and mutants were imaged under reversible hypo-osmotic conditions. For this, the medium was replaced with hypotonic swelling medium (25 mM HEPES in serum free medium) about 5 min prior to imaging. The cells were then imaged for approximately 1 h. Cell osmotic swelling is necessary to eliminate the numerous wrinkles present in the plasma membrane of live cells, which hinder measurement of receptor concentration in the plasma membrane. FRET efficiencies were measured in the plasma membrane of cells not in contact with other cells.

A two-photon microscope equipped with the OptiMis True Line Spectral Imaging system (Aurora Spectral Technologies) was used to image the cells, and a MaiTai laser (Spectra-Physics) was used to generate femtosecond mode-locked pulses[79]. Two images were acquired for each cell: one upon excitation at 840 nm to mainly excite the donor, and a second one at 960 nm to primarily excite the acceptor[43]. The FSI-FRET method was used to measure the FRET efficiency and the concentrations of donor (EphA2-mTurquoise) and acceptor (EphA2-EYFP) in small cell plasma membrane areas. Calibration solutions of purified soluble EYFP and mTurquoise, produced using a published protocol, were used to convert pixel-level intensities of the images into EphA2 concentrations as previously described[43].

The FRET efficiency due to EphA2 dimerization depends on the fraction of receptors that are dimeric, $f_D$, and on the acceptor fraction, $x_A$, according to

$$FRET = f_D x_A \tilde{E} \qquad (1)$$

The Ẽ in Eq. (1) is a constant called the intrinsic FRET, which represents the FRET efficiency in a dimer containing a donor molecule and an acceptor molecule. Intrinsic FRET is a structural parameter that depends only on the separation and orientation of the two fluorescent proteins in the dimer, and not on the dimerization propensity.

The following equation, which provides a link between the experimentally derived parameters $FRET$, $x_A$, total EphA2 concentration $[T]$ (including the concentrations of both EphA2-mTurquoise and EphA2-EYFP), and the two unknowns, $K_{diss}$ and Ẽ, was used to fit the data with MATLAB[43].

$$\frac{FRET}{x_A} = \frac{1}{[T]}\left([T] - \frac{K_{diss}}{4}\left(\sqrt{1 + 8[T]/K_{diss}} - 1\right)\right)\tilde{E} \qquad (2)$$

**FIF imaging and analysis.** HEK293T cells were cultured and seeded as for FRET. The cells were transiently transfected with EphA2 5A-EYFP or EphA2 5E-EYFP plasmids and serum starved overnight. About 5 min before imaging, the cell starvation medium was replaced with hypotonic swelling medium (25 mM HEPES in 25% serum free medium, 75% H$_2$O) and the cells were treated with 5.5 nM ephrinA1-Fc. Images of the plasma membrane in contact with the plate surface were collected with a confocal microscope (Leica TCS SP8) using the photon counting capabilities of the HyD hybrid detector. Measurements were performed with a 488 nm excitation diode laser and the emission spectra of EYFP were collected from 520 to 580 nm at a scanning speed of 20 Hz, pixel depth of 12-bits, image size of 1024×1024 pixels, and a zoom factor of 2.

Images were analyzed using previously published FIF software[46]. The cells were outlined by hand, and the software segmented the outlined region into 15×15 pixel regions of interest. After segmentation, the data were analyzed using the brightness and concentration calculator in the FIF software, which determines the molecular brightness and receptor concentration in each segment[46]. For molecular brightness

calculations, the following equation was used:

$$\varepsilon = \frac{\sigma^2 - \sigma_D^2}{<I>} \qquad (3)$$

where $\sigma^2$ is the variance of fluorescence across segments, $\sigma_D^2$ is the variance of the noise of the detector, and $<I>$ is the average fluorescence intensity. For a photon-counting detector used in our experiments, the brightness is[80]:

$$\varepsilon = \frac{\sigma^2}{<I>} - 1 \qquad (4)$$

The brightness values were calculated for thousands of regions of interest. The distributions of the brightness values are shown in Fig. 6b–e.

**Reporting summary**. Further information on research design is available in the Nature Research Reporting Summary linked to this article.

## Data availability

Atomic structures and diffraction data generated in this study have been deposited in the Protein Data Bank (PDB) under accession codes 7KJA (EphA2 WT), 7KJB (EphA2 S897E/S901E) and 7KJC (EphA2 S901E). HDX-MS data have been deposited in the Mass Spectrometry Interactive Virtual Environment (MassIVE, massive.ucsd.edu) database with accession code MSV000086658. Kinase screening data are presented in Supplementary Table 3. All other data supporting the conclusions are available in the article and in the Source Data file provided with this paper. This study uses publicly available data from PhosphoSite Plus (https://www.phosphosite.org) and the PDB under accession codes: 2QOC, 3KKA, 4PDO, 5EK7, 6FNG and 6Q7D. Source data are provided with this paper.

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

## Acknowledgements

The authors thank P. Hornbeck (PhosphoSitePlus) for providing original data on phosphosites in individual peptides from the PhosphoSite database, Alejandro Conde-Perez for generating the MEL-JUSO-EphA2 stable cell line, B. Emerling for providing the SW626 cell line, P. Itkin-Ansari for the BxPC3 cell line, F. Levine for the PANC1 pancreatic cancer cell line, K. Vuori for the SKOV3 and MDA-MB-468 cell lines, J. Smith for the MDA-MB-231 cell line, R. Maki for the BT549 cell line, W. Stallcup for the U87, T98 and U251-MG cell lines, the staff of the BioCAT Advanced SAXS Training Course, in particular W. Shang, for support with initial SAXS data collection and analysis, staff of the Australian Synchrotron SAXS/WAXS beamline (part of ANSTO) for technical assistance, J. Murphy (WEHI) for help with SAXS data collection and analysis, A. Bobkov for collecting and analyzing AUC data, S. Silletti, UCSD Biomolecular and Proteomics Mass Spectrometry (BPSM) Facility, for HDX-MS data collection and analysis and Ryan Lumpkin for help with DECA. This work was supported by NIH grants GM131374 and AG062617 and institutional funds to EBP, NIH grant GM068619 to KH, and National Cancer Institute Cancer Center Support grant P30 CA030199, which supported SBP Core Facilities and funds for a pilot project. The UCSD BPSM Facility, which carried out the HDX experiments, is supported by the NIH shared instrumentation grant number S10 OD016234 (Synapt-HDX-MS). This project also used resources of the Advanced Photon Source, a U.S. Department of Energy (DOE) Office of Science User Facility operated for the DOE Office of Science by Argonne National Laboratory under Contract No. DE-AC02-06CH11357. The work for this project was supported by grant 9 P41 GM103622 from the National Institute of General Medical Sciences of the National Institutes of Health. Use of the BioCAT Pilatus 3 1 M detector was provided by grant 1S10OD018090-01 from NIGMS. The content is solely the responsibility of the authors and does not necessarily reflect the official views of the National Institute of General Medical Sciences or the National Institutes of Health.

## Author contributions

B.C.L. and E.B.P. designed the overall study and interpreted the results; T.P.L. and K.H. designed and interpreted the FRET and FIF experiments; B.C.L., M.P.G., T.P.L. and M.W.M. performed experiments; C.R.H. helped with SAXS data collection and analysis; B.C.L. and E.B.P. wrote the manuscript with input from T.P.L. and K.H.; E.B.P. and K.H. secured funding.

## Competing interests

The authors declare no competing interests.
