## [Peer Review File · Nature Communications]

Regulation of the EphA2 Receptor Intracellular Region by Phosphomimetic Negative Charges in the Kinase-SAM LinkerREVIEWER COMMENTS

Reviewer #1 (Remarks to the Author):

The work by Lechtenberg et al. “Allosteric regulation of the EphA2 Receptor Intracellular Region by Serine/Threonine Kinases” seeks to uncover the effects of sequential phosphorylation of a linker segment connecting the EphA2 kinase and sterile alpha motif (SAM) domains. These phosphorylation events are important in regulating the structure and dynamics of EphA2, an important enzyme that promotes malignancy. The combination of X-ray crystallography, SAXS, FRET and HDX-MS makes for an in-depth structural and dynamics analysis. However, the manuscript in its present form has some missing links in interpretation that need to be addressed. I believe some of the missing links are due to an oversimplification- that substitution with glutamates mimic the effects of phosphorylation. The authors need to offer more supportive evidence that the functional effects of phosphorylation of EphA2 are equivalent to the glutamate- point mutants. Could this be the reason for why S897E showing no differences in deuterium exchange relative to WT (Fig. 5D)?

For concluding allosteric changes, the authors need to demonstrate changes at sites distal to the kinase: SAM interface. None of these are shown or described by the authors. The authors conclusion of phosphorylation of the linker domain being the basis for allosteric regulation of EphA2 is unsubstantiated by the lack of a description of changes in catalytic activity or distal effects.

Major Comments:

- 1) The two molecules in the asymmetric unit in Figure 1C, E and 1D, F are distinctly different. 1C and 1E show a more compact conformation or as the authors prefer to state- ‘configuration’. The authors’ assessment of the effects of linker phosphorylation is entirely based on the configuration of the molecule in Fig 1C, E. What about the loosely held 1D, F. Based on these results, it would seem that unphosphorylated Eph2A assumes an ensemble of conformations in solution. Phosphorylation is less likely to be a binary switch. This ensemble behavior is not considered in the model presented in Figure S9. The authors also need to discuss if the two conformations in the crystal are artifacts of crystallization. It doesn’t appear so, since SAXS in Figure 3 clearly reveals a more extended ensemble average for WT and S897E/S901E. Strangely, the SAXS profiles in gray are largely superimposable. Any differences are minimal and not readily discernible. A reader is more inclined to believe the SAXS map rather than the overlaid model.
- 2) There is no discussion of dimer formation in the model in Figure S9. Is there any dimerization in the mutants? The authors’ FRET indicates phosphorylation promotes modest dimerization.
- 3) I have a problem with the conclusion- “Hydrogen-deuterium exchange reveals that the EphA2 kinase domain interacts with the SAM domain” HDXMS is excellent for mapping changes. However, it is difficult

to correlate intraprotein interaction interfaces from deuterium exchange of a single protein alone. There may be multiple surface peptides showing decreased exchange, despite being highly solvent accessible. How certain are the authors that the [G-I] region alone is the locus for interactions with SAM domain. That the authors have a crystallography structural snapshot of the kinase:SAM domain interface is another matter. This conclusion needs to be rewritten.

A reference needs to be provided by the authors that describes that deuterium exchange (<5 min) preferentially reports changes in solvent accessibility (Peacock, Davis, Markwick and Komives, Biochemistry (2019)). Notwithstanding the reference, the authors should describe why their deuterium exchange only reports on solvent accessibility and not H-bonding propensities.

4) The title and abstract need to be reworded. There is no evidence for canonical phosphorylation in this article.

This manuscript has extensive orthogonal data sets: crystallographic analysis, mutagenesis, SAXS, FRET and HDXMS. All the methods are described in satisfactory detail. However, these have not all been integrated into one cohesive interpretive model. That and the assumption that glutamates can precisely mimic the effects of phosphorylation form the major weakness of this manuscript.

Reviewer #2 (Remarks to the Author):

The paper by Pasquale and colleagues presents for the first time the full length structure of the intracellular region of an Eph receptor protein; before the C-terminal SAM domain was found to be invisible, possibly degraded. Furthermore this is the first time amide hydrogen exchange is applied to this region of any Eph receptor to measure the local stability of the structure/accessibility of amides to solvent. The results are highly significant for this well focused project on Ser/Thr phosphorylation of the Kinase-SAM domain linker region, as the data suggest the system exists in a closed and an open state, which would explain previously observed autoinhibition of EphA2 kinase activity caused by the presence of the SAM domain and substantiate the different mechanism which operates for this receptor in non-canonical signaling. The paper is overall well executed and written but there are several serious gaps which need to be filled in order for the conclusions to be even more compelling:

1) SAXS experiments need to be carried out with the 5E and 5A, E820K/E825K mutants, as the other experiments did not detect very significant changes for the single/double S to E mutants.

2) The authors do not consider that there could be a dimeric form in their crystal structures and the contacts between linker region and SAM/Kinase domain are not explored by mutagenesis. A PISA

analysis should be carried out to evaluate whether the KD-SAM interfaces are substantial. Similarly, significant crystal contacts need to be mentioned - to what extent is the crystal structure physiologically relevant? (re. dimerization, the protein concentration used in the analytical ultracentrifugation experiment, 9 μ M is many-fold lower than used for SAXS; please comment).

3) It is a bit confusing for which functional aspect of non-canonical signaling an allosteric network between the linker, SAM and JM region may be required. Unless this can be tested experimentally, such aspects should be dropped.

4) A figure similar to S9 should be in the main paper, but the figure shows a number of aspects of the system which are not investigated here (and have been inferred from simulations, e.g. the kinase-domain membrane interactions, rather than from experiments). Specifically the persistence of the same(?) contacts between SAM and active kinase domain in canonical signaling is at odds with the observation that the presence of the SAM domain is kinase autoinhibitory. It would be better to have a model focusing on what this work found, rather than be overly speculative.

Minor issues:

4. For discussion: is EphA2 phosphorylation on Ser/Thr outside the linker region?

5. The paper suggests that it has not yet been investigated/discussed whether noncanonical vs. noncanonical Tyr vs. Ser/Thr phosphorylation mechanism are mutually exclusive. This is not quite accurate - the authors themselves suggested this earlier but data from other laboratories suggest that this may not be the case or at least is rather cell dependent.

6. Ref. 39 should be mentioned earlier in the paper, not just the discussion. e.g. at line 33 would be an appropriate place.

7. The last line of the abstract - that these results can inform new therapeutic strategies- is not elaborated anywhere but should be discussed or deleted.

8. line 228 - likely absence of allosteric coupling because the linker is relatively unstructured- seems to contradict the discussion and possible coupling by longer range contacts.

Reviewer #3 (Remarks to the Author):

The manuscript by Lechtenberg and colleagues reports the crystal structure of the complete intracellular region of EphA2, as well as additional small-angle X-ray scattering, hydrogen-deuterium exchange mass spectrometry, cell-based FRET and phosphorylation assays, all aimed at gaining insight into the structure and dynamics of the EphA2 intracellular region. The research integrates multiple diverse techniques and approaches, but a major problem is that the experiments are significantly overinterpreted and do not adequately support the proposed model or conclusions.

Major Points:

1. The authors claim that their experiments support the existence of two conformations of the EphA2 intracellular region, “open” and “closed”. This, though, is not really the case. The crystallographic and small-angle X-ray scattering structural data in fact point to the kinase and SAM domains being two globular domains connected by a flexible linker regardless of the phosphorylation status. Indeed, this is exactly what molecule B on Fig. 1 (WT), the structure on Fig. 2 (S897E/S901E), as well as the structure of the S901E mutant, represent (just different crystal packing). The more compact structure of molecule A on Fig. 1 is probably also due to different crystal packing and if the authors claim that it represents a biologically relevant “closed” conformation, they need to perform structure-based mutagenesis studies to show that the observed here Kinase/SAM interface residues (R857, R860, E914, E911, E857) indeed affects the biological (or even biophysical) properties of EphA2. The small-angle X-ray scattering data on Fig. 3 further show that WT and S897E/S901E EphA2 have exactly the same structure.

2. While the hydrogen-deuterium exchange experiments are consistent with the negatively charged kinase domain region (around residues 810-830) interacting with the C-terminal half of the SAM domain, they do not suggest or prove the existence or relevance of such an interaction. These are indeed the EphA2 surface regions with the most extreme surface electrostatic potentials and the hydrogen-deuterium exchange differences (Fig. 4A) might just be due to the biophysical properties of these regions containing expansive highly charged surfaces. Furthermore the proposed here “closed” conformation resulting from the interaction of these surface regions is completely different from the more compact (“closed”?) structure of molecule A in the crystal structure on Fig. 1. Indeed in none of the crystal structures, both in the asymmetric unit interactions and the crystallographic packing interactions, such kinase (810-830)-SAM(C-terminal half) contacts are reported. One would assume that such kinase (810-830)-SAM(C-terminal half) contacts could form both intra-molecularly and inter-molecularly in the crystals (including crystal packing) if they are strong enough to be biologically relevant.

3. Further regarding the hydrogen-deuterium exchange experiments, one would assume that direct electrostatic disruption of the (810-830)-SAM(C-terminal half) interactions, if they indeed exist, would affect the hydrogen-deuterium exchanges much more than indirect disruption via mutations in the linker region; this, though, is not the case (compare Fig. 5A to Fig. 5F in the SAM region). In addition, the linker region mutations seem to similarly affect the hydrogen-deuterium exchanges in the N-terminal and C-terminal SAM regions (Fig. 5), which is not consistent with only the C-terminal region participating in the Kinase-SAM interactions (as suggested by Fig. 4).

5. How do the authors explain the observed “stabilization of the kinase-SAM linker” in the mutations that are supposed to promote the “open” conformation (lines 262-265), when the intuitive expectation would be that an “open” conformation would have a less stable and more flexible linker?

4. If the authors claim that the (810-830)-SAM(C-terminal half) interactions are central to the formation of the “closed” conformation, why were mutations directly disrupting these electrostatic interactions (e.g. E820K/E825K) not tested in the FSI-FRET cell-based assays (lines 280-285)? Note that such interactions, if indeed existing, could possibly form both intra-molecularly and inter-molecularly on the cell surface.

5. The authors use the terms “allosteric regulation” and “allosteric regulatory network” throughout the manuscript, title, discussion, and proposed model, but their data does not directly show any allostery in any of their experiments. The closest link to possible allosteric effects I could find is an observation of a small difference in the structure of a surface loop (lines 161-162, Fig. 2B) that has been previously proposed to have a possible role in an allosteric network in another receptor (EphA3)?

6. Lines 245-249: “Surprisingly, the effect of a single negative charge at position 897, the functionally best characterized linker phosphorylation site, does not produce a particularly strong effect on hydrogen-deuterium exchange in these regions (Fig. 5E), suggesting that phosphorylation of S897 alone is not sufficient to regulate EphA2 signaling properties.” Not really. What this more likely suggests is that the hydrogen-deuterium exchange changes in these regions may not adequately (or fully) represent the regulation of the EphA2 signaling properties via S897 phosphorylation.

Minor points:

1. The authors state that in vivo a maximum of 3 phosphorylated residues in the Kinase-SAM linker region are observed (line 91), yet in order to get significant effects in their biophysical studies they have to mutate all 5 potential phosphorylation sites. Is mutating all 5 sites reflective of a biologically relevant state?

2. Lines 131-133, “The conformational flexibility of the S901 motif also supports the previously proposed idea that the negative charge introduced by S897 phosphorylation, and not a specific conformational change induced by S897 phosphorylation, is critical for subsequent phosphorylation of S901 by CK1 acidophilic kinases”. Not clear how it supports it?

Reviewer #1

The work by Lechtenberg et al. “Allosteric regulation of the EphA2 Receptor Intracellular Region by Serine/Threonine Kinases” seeks to uncover the effects of sequential phosphorylation of a linker segment connecting the EphA2 kinase and sterile alpha motif (SAM) domains. These phosphorylation events are important in regulating the structure and dynamics of EphA2, an important enzyme that promotes malignancy. The combination of X-ray crystallography, SAXS, FRET and HDX-MS makes for an in-depth structural and dynamics analysis. However, the manuscript in its present form has some missing links in interpretation that need to be addressed. I believe some of the missing links are due to an oversimplification- that substitution with glutamates mimic the effects of phosphorylation. The authors need to offer more supportive evidence that the functional effects of phosphorylation of EphA2 are equivalent to the glutamate-point mutants. Could this be the reason for why S897E showing no differences in deuterium exchange relative to WT (Fig. 5D)?

Reply. In many instances in the literature, glutamate has been used to mimic a phosphorylated residue. When the effects of phosphorylation are due to the negative charge of phosphate, glutamate is very often capable of at least in part functionally mimicking a phosphorylated residue (as exemplified by the studies reported in many articles, including ^{1, 2, 3, 4, 5, 6, 7, 8}). In fact, for 70% to 90% of ~700 human phosphosites with demonstrated functional importance that were analyzed by mutagenesis, serine/threonine replacement with glutamic/aspartic acid mimicked the expected functional effects of phosphorylation (Table S3A in⁹). Furthermore, comparative genomic analyses suggest that ~5% of phosphosites may have evolved from glutamic or aspartic acid, and there is also evidence of the opposite evolution from phosphorylated residues to negatively charged residues⁵.

Bulk electrostatic effects seem to be an important mechanism through which clusters of phosphorylated residues in unstructured regions, such as linkers, can affect protein function, particularly when the number of the phosphosites appears to be more critical than their exact positions^{10, 11, 12, 13}. This seems to be the case for the EphA2 kinase-SAM linker, where many different combinations of 2 and 3 of the 5 possible phosphorylation sites have been detected, as mentioned in the Results section (pages 3 and 10) and shown in the Supplementary Fig. 1g. These variable patterns of linker phosphorylation also make it unlikely that the role of the phosphosites is to bind specific protein domains recognizing phosphorylated motifs (such as WW, Forkhead, WD40 or 14-3-3 domains), a role that would not be mimicked by glutamic or aspartic acid.

For concluding allosteric changes, the authors need to demonstrate changes at sites distal to the kinase:SAM interface. None of these are shown or described by the authors. The authors conclusion of phosphorylation of the linker domain being the basis for allosteric regulation of EphA2 is unsubstantiated by the lack of a description of changes in catalytic activity or distal effects.

Reply. The hypothesized interface between kinase and SAM domains is distal to the linker, and we also observe small but consistent changes in HDX in the juxtamembrane region as a consequence of linker mutations. This is why we considered the effect of linker phosphorylation allosteric, according to literature on linkers including (1) “Dynamic allostery: linkers are not merely flexible”¹⁴; (2) “The role of protein loops and linkers in conformational dynamics and allostery”¹⁵; and (3) “A strategy to identify linker-based modules for the allosteric regulation of antibody-antigen binding affinities of different scFvs”¹⁶). However, since we are not showing changes in EphA2 enzymatic activity, we have removed mention of allosteric effects, in keeping with the more widely accepted definition of allostery.

Major Comments:

1) The two molecules in the asymmetric unit in Figure 1C, E and 1D, F are distinctly different. 1C and 1E show a more compact conformation or as the authors prefer to state- ‘configuration’. The

authors' assessment of the effects of linker phosphorylation is entirely based on the configuration of the molecule in Fig 1C, E. What about the loosely held 1D, F. Based on these results, it would seem that unphosphorylated Eph2A assumes an ensemble of conformations in solution. Phosphorylation is less likely to be a binary switch. This ensemble behavior is not considered in the model presented in Figure S9. The authors also need to discuss if the two conformations in the crystal are artifacts of crystallization. It doesn't appear so, since SAXS in Figure 3 clearly reveals a more extended ensemble average for WT and S897E/S901E. Strangely, the SAXS profiles in gray are largely superimposable. Any differences are minimal and not readily discernible. A reader is more inclined to believe the SAXS map rather than the overlaid model.

Reply. We agree that unphosphorylated EphA2 may assume an ensemble of conformations in solution and that phosphorylation is unlikely to be a binary switch. Rather, phosphorylation could shift the equilibrium towards an ensemble of more open conformations, particularly since the number of linker phosphosites can vary. We have modified the Abstract, the text of the revised manuscript and the model in Fig. S9 (Fig. 8 in the revised manuscript) to better convey this dynamic view.

We would also like to note that crystallization selects for one or several stable conformations among the multiple conformations a protein may assume in solution, as required for the protein to pack into the crystal lattice. This is especially expected for a system like the EphA2 intracellular region, consisting of two folded domains connected by a flexible linker. Based on only the crystal structure, it is difficult to establish how well the conformations in the crystal reflect major conformations in solution. This is why we used SEC-SAXS and HDX-MS to investigate EphA2 conformations in solution.

To address the concern of this reviewer, we have performed additional SAXS experiments to analyze multiple EphA2 mutants (S892E, S897E, 3E and 5E) in comparison to EphA2 WT, using the same portion of the EphA2 intracellular region also used for the HDX experiments (S570-I976 instead of D590-I976 used in the previous SAXS experiments). The new SAXS data reveal a clear difference in solution between EphA2 WT and the 5E mutant as well as, to a smaller extent, the S892E mutant (new Fig. 3), in agreement with our observations using HDX-MS. These new SAXS data confirm that of the 3 molecules in our crystal structures (WT molecule A, WT molecule B and S897E/S901E), WT molecule B most closely matches the observed SAXS envelope for EphA2 WT (Fig. 3d). Importantly, the new SAXS data for the EphA2 5E mutant point to a more elongated molecule more similar to the S897E/S901E crystal structure, supporting our hypothesis that negative charges in the linker shift the equilibrium towards more open EphA2 conformations. The SAXS data for EphA2 S892E, but not the other glutamic acid mutants, also show a small tendency towards a larger radius, potentially suggesting a particularly important role for S892 phosphorylation. We now discuss this on page 5 of the revised manuscript.

2) There is no discussion of dimer formation in the model in Figure S9. Is there any dimerization in the mutants? The authors' FRET indicates phosphorylation promotes modest dimerization.

Reply. Our FRET data in Fig. S8 show that both EphA2 WT and the various linker mutants can dimerize when their density in the plasma membrane is high. Linker phosphorylation seems to slightly promote dimerization since K_{diss} for the EphA2 5A mutant, which cannot be phosphorylated, is ~3-fold higher than for EphA2 WT (in which the linker is partially phosphorylated) and ~4-fold higher than for the EphA2 5E mutant (which has 5 negatively charged residues in the linker replacing all linker residues that can be phosphorylated). New exciting fluorescence intensity fluctuation (FIF) experiments show that the negative charges in the EphA2 5E mutant enable formation of large EphA2 clusters in cells stimulated with the ephrinA1-Fc ligand while the EphA2 5A mutant forms smaller oligomers. We have included oligomerization in the model shown in Fig. S9 (Fig. 8 in the revised manuscript) to reflect these findings and the new FIF data we obtained (Fig. 6 in the revised manuscript).

3) I have a problem with the conclusion- “Hydrogen-deuterium exchange reveals that the EphA2 kinase domain interacts with the SAM domain” HDXMS is excellent for mapping changes. However, it is difficult to correlate intraprotein interaction interfaces from deuterium exchange of a single protein alone. There may be multiple surface peptides showing decreased exchange, despite being highly solvent accessible. How certain are the authors that the α G- α I region alone is the locus for interactions with SAM domain. That the authors have a crystallography structural snapshot of the kinase:SAM domain interface is another matter. This conclusion needs to be rewritten. A reference needs to be provided by the authors that describes that deuterium exchange (<5 min) preferentially reports changes in solvent accessibility (Peacock, Davis, Markwick and Komives, *Biochemistry* (2019)). Notwithstanding the reference, the authors should describe why their deuterium exchange only reports on solvent accessibility and not H-bonding propensities.

Reply. We have included on page 5 what we believe was the suggested reference (Markwick et al. 2019 *Biophys J* ¹⁷), which supports the notion that changes in hydrogen/deuterium exchange in the time scale of a few minutes preferentially report changes in solvent accessibility, as we propose, rather than changes in hydrogen bonding. Regarding the proposed interaction between the kinase and SAM domains, we admit that our original title “Hydrogen-deuterium exchange reveals that the EphA2 kinase domain interacts with the SAM domain” may have been worded too strongly. In the revised manuscript, we have changed this title to: “Hydrogen-deuterium exchange supports an interaction between the EphA2 kinase and SAM domains” to better reflect that this is a potential interpretation of our data. However, we believe that this is the most likely interpretation for multiple reasons:

- a) As mentioned by this reviewer, we measured the exchange in the “fast limit” (a few minutes), which preferentially reports on solvent accessibility of amide protons not engaged in stable hydrogen bonding.
- b) We observed correlated hydrogen-deuterium exchange rates in two regions in different domains: the kinase and SAM domains. These two domains are connected by a flexible linker and thus it seems unlikely that these correlated exchange processes are just a coincidence.
- c) The two regions show similar hydrogen-deuterium exchanges not only in EphA2 WT (Fig. 4a), but also when we introduce mutations in the linker (Fig. 5). Again, it seems very unlikely that these correlated changes in hydrogen-deuterium exchange are due to independent changes in hydrogen bonding in each of the two domains, especially since the mutations introduced are in the flexible linker connecting the domains rather than within the domains.
- d) The two regions have opposite charges (Fig. 4c), consistent with their potential interaction.
- e) As discussed below in the reply to point 2 of Reviewer 2, we observe intermolecular interactions between the kinase domain and the SAM domain in our EphA2 WT crystal structure. We included a new Fig. 4d to highlight this interaction and a brief section in the Results describing it.

Overall, an effect of negative charges in the linker on the interaction between the EphA2 kinase and SAM domains seems a far more likely explanation of the observed changes in hydrogen-deuterium exchange than effects on the hydrogen bonding propensities of residues in two separate domains, neither of which includes the mutated residues. Thus, in the manuscript we still present what we believe is the most parsimonious explanation of our data. However, to address the concern of the reviewer, we have also modified the text on page 10 in the Discussion to state the possibility (in our opinion remote) that the observed changes in hydrogen/deuterium exchange might reflect changes in hydrogen-bonding propensities rather than changes in solvent accessibility.

4) The title and abstract need to be reworded. There is no evidence for canonical phosphorylation in this article.

Reply. We assume that this comment refers to EphA2 non-canonical phosphorylation (i.e. phosphorylation of the linker). We have modified the title and the abstract to indicate that we have evaluated the effects of negative charges introduced by mutation to mimic the negative charges introduced by phosphorylation.

5) This manuscript has extensive orthogonal data sets: crystallographic analysis, mutagenesis, SAXS, FRET and HDX-MS. All the methods are described in satisfactory detail. However, these have not all been integrated into one cohesive interpretive model. That and the assumption that glutamates can precisely mimic the effects of phosphorylation form the major weakness of this manuscript.

Reply. We have now performed new SAXS experiments that investigate most of the EphA2 mutants analyzed in the HDX-MS experiments. This allows us to better integrate the different types of data into a cohesive model, which is shown in the revised Fig. S9 (Fig. 8 in the revised manuscript). We also tried to better correlate our multiple orthogonal datasets throughout the text. Our model does not assume that glutamate can precisely mimic all effects of phosphorylation. It is well known, as we have also published¹⁸, that mutation of a phosphorylated residue to glutamate does not recreate a phosphorylated binding motif, such as those recognized by SH2, PTB, WW, Forkhead, WD40, 14-3-3 etc. domains. In addition, the charge of a phosphate group and its ionic shell are somewhat different from those of glutamate¹⁹. However, as mentioned above in the reply to the first general comment of this reviewer, many studies have shown that residues with negatively charged side chains can functionally mimic phosphorylated residues when the negative charge determines function. In any case, we have modified the manuscript to more carefully distinguish negatively charged from phosphorylated amino acids.

Reviewer #2

The paper by Pasquale and colleagues presents for the first time the full length structure of the intracellular region of an Eph receptor protein; before the C-terminal SAM domain was found to be invisible, possibly degraded. Furthermore this is the first time amide hydrogen exchange is applied to this region of any Eph receptor to measure the local stability of the structure/accessibility of amides to solvent. The results are highly significant for this well focused project on Ser/Thr phosphorylation of the Kinase-SAM domain linker region, as the data suggest the system exists in a closed and an open state, which would explain previously observed autoinhibition of EphA2 kinase activity caused by the presence of the SAM domain and substantiate the different mechanism which operates for this receptor in non-canonical signaling. The paper is overall well executed and written but there are several serious gaps which need to be filled in order for the conclusions to be even more compelling:

1) SAXS experiments need to be carried out with the 5E and 5A, E820K/E825K mutants, as the other experiments did not detect very significant changes for the single/double S to E mutants.

Reply. We now include SAXS experiments for EphA2 WT and the S892E, S897E, 3E, and 5E EphA2 mutants also used in the HDX-MS experiments. In these experiments, we observe small differences between EphA2 WT and S892E and pronounced differences between EphA2 WT and the 5E mutant. The SAXS profile of the EphA2 5E mutant indicates a much more elongated molecule (most evident in the distance distribution plots in revised Fig. 3b), supporting our hypothesis that introducing negative charges (mimicking phosphorylation) in the linker favors more open states of the EphA2 intracellular region.

2) The authors do not consider that there could be a dimeric form in their crystal structures and the contacts between linker region and SAM/Kinase domain are not explored by mutagenesis. A PISA

analysis should be carried out to evaluate whether the KD-SAM interfaces are substantial. Similarly, significant crystal contacts need to be mentioned - to what extent is the crystal structure physiologically relevant? (re. dimerization, the protein concentration used in the analytical ultracentrifugation experiment, 9 μM is many-fold lower than used for SAXS; please comment).

Reply. We previously characterized the oligomerization state of the EphA2 WT kinase-SAM construct by analytical ultracentrifugation and observed that under these conditions ($\sim 9 \mu\text{M}$, 0.45 mg/ml EphA2 intracellular region in 10 mM HEPES pH 7.9, 100 mM NaCl at 20°C), the protein is monomeric in solution (Fig. S2 in the original manuscript, now Fig. S2a in the revised manuscript).

A

PISA Interface List.

Session Map (id=436-81-M4K)

Start **Interfaces** Interface Search

Monomers

Assemblies

Interfaces in PDB 7kja crystal.

Space symmetry group: I 1 2 1. Resolution: 1.75 Å

CRYSTAL STRUCTURE OF THE EPHA2 INTRACELLULAR KD-SAM DOMAINS

Interfaces XML View Details Download Search

##	NN	Structure 1			x	Structure 2			interface		N _{IP}	N _{SB}	N _{DS}	CSS					
		Range	N _{at}	N _{DR}		Surface A ²	Range	Symmetry op-n	Sym.ID	N _{at}					N _{DR}	Surface A ²	area, A ²	ΔG , kcal/mol	ΔG , P-value
1	●	A	69	17	19192	◇	B	-x+1/2,y-1/2,-z-1/2	4_544	77	21	18712	688.5	-4.1	0.413	3	2	0	0.000
2	○	B	72	18	18712	◇	A	-x,y,-z	2_555	75	19	19192	683.0	-4.3	0.375	3	1	0	0.000
3	○	B	80	22	18712	◇	A	x,y,z	1_555	63	15	19192	637.1	-7.3	0.190	6	4	0	0.000
4	○	A	57	18	19192	◇	B	x-1/2,y+1/2,z-1/2	3_454	52	13	18712	542.7	-3.3	0.252	1	4	0	0.000
5	○	B	52	13	18712	×	B	-x+1/2,y-1/2,-z-1/2	4_544	55	19	18712	487.9	0.5	0.662	1	0	0	0.000
6	○	A	54	14	19192	×	A	-x-1/2,y-1/2,-z-1/2	4_444	51	15	19192	468.8	-2.9	0.466	1	1	0	0.000
7	○	[ACP]B:1003	31	1	611	f	B	x,y,z	1_555	53	24	18712	395.6	-1.8	0.520	3	0	0	0.060
8	○	[ACP]A:1002	31	1	613	f	A	x,y,z	1_555	53	23	19192	394.0	-0.2	0.640	2	0	0	0.009
9	○	A	26	12	19192	◇	A	-x,y,-z	2_555	26	12	19192	260.0	1.4	0.778	0	0	0	0.000
10	○	B	22	8	18712	◇	B	-x+1,y,-z	2_655	22	8	18712	213.7	-0.4	0.576	0	2	0	0.000
11	○	B	27	12	18712	◇	A	-x+1/2,y-1/2,-z-1/2	4_544	20	7	19192	210.5	-2.2	0.314	1	0	0	0.000
12	○	B	16	7	18712	×	B	x,y-1,z	1_545	15	5	18712	149.0	1.1	0.686	0	4	0	0.000
13	○	[GOL]A:1001	6	1	215	f	A	x,y,z	1_555	22	7	19192	129.8	0.2	0.658	1	0	0	0.002
14	○	[GOL]B:1002	5	1	218	f	B	x,y,z	1_555	21	7	18712	119.1	-0.2	0.561	1	0	0	0.013
15	○	[GOL]B:1004	6	1	214	f	B	x,y,z	1_555	17	5	18712	100.3	-0.7	0.484	0	0	0	0.014
16	○	[GOL]B:1001	5	1	217	f	B	x,y,z	1_555	13	5	18712	92.4	0.2	0.587	2	0	0	0.014
17	○	[GOL]A:1003	6	1	216	◇	B	-x,y,-z	2_555	9	3	18712	90.5	-0.4	0.547	0	0	0	0.000
18	○	[GOL]A:1003	4	1	216	f	A	x,y,z	1_555	13	5	19192	69.9	0.0	0.585	1	0	0	0.003
19	○	[GOL]B:1004	6	1	214	◇	A	x,y,z	1_555	10	2	19192	58.9	-0.5	0.507	0	0	0	0.000
20	○	[MG]A:1004	1	1	98	f	A	x,y,z	1_555	8	4	19192	43.9	-4.4	0.000	0	0	0	0.038
21	○	B	4	2	18712	◇	A	-x,y-1,-z	2_545	6	3	19192	37.7	-0.5	0.363	0	0	0	0.000
22	○	[MG]A:1004	1	1	98	f	[ACP]A:1002	x,y,z	1_555	5	1	613	37.6	-5.4	0.000	0	0	0	0.047
23	○	B	1	1	18712	×	B	-x+1,y-1,-z	2_645	1	1	18712	14.9	0.7	0.905	0	0	0	0.000

View Details Download Search

B

PISA Assembly List.

Session Map (id=436-81-M4K)

Start **Interfaces** Interface Search

Monomers

Assemblies

Probable Assemblies in PDB 7kja crystal.

Space symmetry group: I 1 2 1. Resolution: 1.75 Å

CRYSTAL STRUCTURE OF THE EPHA2 INTRACELLULAR KD-SAM DOMAINS

Complex Analysis of the complex represented by the coordinate section only of the PDB entry.

Analysis of the protein interfaces has not revealed any specific interactions that could result in the formation of stable quaternary structures. Most probably, the structures do not form a complex in solution.

7kja:A,7kja:B,7kja:[GOL]A:1001,7kja:[GOL]A:1003,7kja:[GOL]B:1001,7kja:[GOL]B:1002,7kja:[GOL]B:1004,7kja:[ACP]A:1002,7kja:[ACP]B:1003,7kja:[MG]A:1004

Fig. 1 for the reviewers. PDBe PISA analysis of our crystal structure of the EphA2 WT intracellular region (PDB ID 7KJA) does not identify any significant stable assemblies. All tested potential assemblies between 2 EphA2 molecules identified in the crystal packing feature a complexation significance score (CSS) of 0.00.

The analytical ultracentrifugation was performed at 9 μM to allow for an optimal signal to noise ratio without overloading the detector. We have now performed additional biophysical characterization of the EphA2 intracellular region in solution using SEC-MALS (size-exclusion chromatography coupled to multi-angle light scattering) at a protein concentration of 3.3 mg/ml ($\sim 70 \mu\text{M}$) (Fig. S2b). This experiment clearly confirms the monomeric nature of the intracellular portion of EphA2 at a concentration similar to that used for SEC-SAXS (2-5 mg/ml). We also performed a PISA analysis as requested by this reviewer. PISA does not find any significant stable assemblies of the EphA2 intracellular region (see Fig. 1 for the reviewers only). Thus, EphA2 dimerization appears to require the full-length receptor and/or its insertion in the plasma membrane.

Given our SEC-MALS data and PISA analysis, we do not believe that the crystal contacts reflect physiologically relevant EphA2 dimerization interfaces. However, we hypothesize that the intermolecular kinase-SAM interaction observed in our structure (and now shown more explicitly in a new Fig. 4d) may mimic a significant intramolecular interaction relevant for EphA2 regulation (see also reply to Reviewer 1, major comment 3).

3) It is a bit confusing for which functional aspect of non-canonical signaling an allosteric network between the linker, SAM and JM region may be required. Unless this can be tested experimentally, such aspects should be dropped.

Reply. We observed clear effects of negative charges in the EphA2 linker on hydrogen-deuterium exchange in the juxtamembrane segment (Fig. 5), which suggest long-range conformational effects. We therefore report this observation on pages 6 and 9-10 of the revised manuscript, although we no longer mention an allosteric network. Given the importance of the juxtamembrane segment in the regulation of EphA2 signaling, it seems important to report these findings. Our data may inform future work and/or help explain observations by other groups.

4) A figure similar to S9 should be in the main paper, but the figure shows a number of aspects of the system which are not investigated here (and have been inferred from simulations, e.g. the kinase-domain membrane interactions, rather than from experiments). Specifically the persistence of the same (?) contacts between SAM and active kinase domain in canonical signaling is at odds with the observation that the presence of the SAM domain is kinase autoinhibitory. It would be better to have a model focusing on what this work found, rather than be overly speculative.

Reply. We have revised the model shown in the original Fig. S9 to only include findings from our manuscript, as suggested, and this is now included as Fig. 8 in the revised manuscript.

Minor issues:

5) For discussion: is EphA2 phosphorylation on Ser/Thr outside the linker region?

Reply. Additional Ser/Thr phosphorylation sites have been identified outside the linker region, including multiple sites in the juxtamembrane segment and a site in the kinase domain (phosphosite.org). We now mention other EphA2 Ser/Thr phosphorylation sites on page 3 of the revised manuscript. However, currently there is no functional information on these sites, and it is not known which kinases phosphorylate them.

6) The paper suggests that it has not yet been investigated/discussed whether noncanonical vs. noncanonical Tyr vs. Ser/Thr phosphorylation mechanisms are mutually exclusive. This is not quite accurate - the authors themselves suggested this earlier but data from other laboratories suggest that this may not be the case or at least is rather cell dependent.

Reply. We have eliminated canonical signaling in the scheme in Fig. 8 (previously Fig. S9), to address a number of reviewers' comments. We have also modified the paragraph near the end of

the Discussion on page 11 to better explain what is known about the relation between canonical and non-canonical signaling.

7) Ref. 39 should be mentioned earlier in the paper, not just the discussion. e.g. at line 33 would be an appropriate place.

Reply. We have included reference 39, as recommended.

8) The last line of the abstract - that these results can inform new therapeutic strategies- is not elaborated anywhere but should be discussed or deleted.

Reply. We have deleted the last line of the Abstract.

9) line 228 - likely absence of allosteric coupling because the linker is relatively unstructured- seems to contradict the discussion and possible coupling by longer range contacts.

Reply. We have eliminated the problematic sentence.

Reviewer #3

The manuscript by Lechtenberg and colleagues reports the crystal structure of the complete intracellular region of EphA2, as well as additional small-angle X-ray scattering, hydrogen-deuterium exchange mass spectrometry, cell-based FRET and phosphorylation assays, all aimed at gaining insight into the structure and dynamics of the EphA2 intracellular region. The research integrates multiple diverse techniques and approaches, but a major problem is that the experiments are significantly overinterpreted and do not adequately support the proposed model or conclusions.

Major Points:

1) The authors claim that their experiments support the existence of two conformations of the EphA2 intracellular region, “open” and “closed”. This, though, is not really the case. The crystallographic and small-angle X-ray scattering structural data in fact point to the kinase and SAM domains being two globular domains connected by a flexible linker regardless of the phosphorylation status. Indeed, this is exactly what molecule B on Fig. 1 (WT), the structure on Fig. 2 (S897E/S901E), as well as the structure of the S901E mutant, represent (just different crystal packing). The more compact structure of molecule A on Fig. 1 is probably also due to different crystal packing and if the authors claim that it represents a biologically relevant “closed” conformation, they need to perform structure-based mutagenesis studies to show that the observed here Kinase/SAM interface residues (R857, R860, E914, E911, E857) indeed affects the biological (or even biophysical) properties of EphA2. The small-angle X-ray scattering data on Fig. 3 further show that WT and S897E/S901E EphA2 have exactly the same structure.

Reply. We now more clearly describe (also in response to suggestions from Reviewer 1) that introduction of negative charges in the EphA2 linker (by linker phosphorylation or, as in our experiments, using phosphomimetic mutations) shifts the equilibrium between more “closed” and more “open” conformational ensembles. We believe that this model is well supported by our HDX-MS and SEC-SAXS data, including our new SEC-SAXS experiments clearly pointing to a more elongated conformation of the EphA2 5E mutant in solution. We have further mutated two residues in the electronegative surface of the kinase domain (in the EphA2 E820K-E825K mutant), which in the HDX-MS experiments produced effects that are similar to those of the 3E (T898-S899E-S901E) and S892E mutations (Fig. 5c,d,f). These results are consistent with our model. We have removed the figure panels showing the detailed interactions between the kinase domain and SAM domains

(previously Fig 1e,f) and reworded the corresponding text in the Results section to de-emphasize these specific interactions.

Of all our crystal structures, EphA2 WT molecule B most closely matches the EphA2 WT SAXS envelope, but we agree with the reviewer that this likely only represents one possible conformation of the ensemble of closed conformations. Our new SAXS data now also show that the 5E mutant adopts an elongated shape in solution, which we interpret as a shift to more “open” conformations. The EphA2 5E SAXS envelope most closely matches the conformation observed in the S897E/S901E crystal structure. We agree that the conformations observed in the crystal structures may not correspond exactly to major conformations in solution since the linker is likely flexible in both the “open” and “closed” states, and each of these states likely includes multiple conformations. This may explain why the SAXS envelopes do not exactly match the conformations observed in the crystal structures. We have updated the text at the end of the section describing the SAXS experiments accordingly.

2) While the hydrogen-deuterium exchange experiments are consistent with the negatively charged kinase domain region (around residues 810-830) interacting with the C-terminal half of the SAM domain, they do not suggest or prove the existence or relevance of such an interaction. These are indeed the EphA2 surface regions with the most extreme surface electrostatic potentials and the hydrogen-deuterium exchange differences (Fig. 4A) might just be due to the biophysical properties of these regions containing expansive highly charged surfaces. Furthermore the proposed here “closed” conformation resulting from the interaction of these surface regions is completely different from the more compact (“closed”?) structure of molecule A in the crystal structure on Fig. 1. Indeed in none of the crystal structures, both in the asymmetric unit interactions and the crystallographic packing interactions, such kinase (810-830)-SAM(C-terminal half) contacts are reported. One would assume that such kinase (810-830)-SAM(C-terminal half) contacts could form both intra-molecularly and inter-molecularly in the crystals (including crystal packing) if they are strong enough to be biologically relevant.

Reply. We have modified the text on page 10 of the Discussion to include the possibility that the time-dependent increase in hydrogen-deuterium exchange in the kinase domain and SAM domain regions shown in Fig. 4A may depend on the biophysical properties of these highly charged surfaces. However, our experiments show that (1) linker mutations, which are outside the kinase domain or the SAM domain, affect in a concerted manner HDX in both domains, and (2) the E820K-E825K mutations in the kinase domain affect HDX in the SAM domain in a manner similar to linker glutamic acid mutations. The most parsimonious explanation for these observations is that the kinase and SAM domain surfaces with opposite charges interact with each other, whereas effects of the linker mutations on the intrinsic biophysical properties of regions in distal domains seem less likely.

Regarding the presence of the proposed interface in the crystal structures, in the EphA2 WT structure we actually observe packing of the positively charged region in the SAM domain of molecule A with the negatively charged region in the kinase domain of molecule B. This interaction is visible in Fig. 1B, although we did not highlight this in the previous version of the manuscript. We now more clearly show this interaction in the new Fig 4d. The fact that we observe the interaction between two EphA2 molecules in the crystal structure, rather than an intramolecular interaction, may be due to the very high protein concentration in the crystal and to the fact that the intermolecular interaction may be more favorable to crystal nucleation and crystal growth. In contrast, the lower EphA2 concentration in the cell would favor an intramolecular interaction. We now explain this in on page 10 of the revised manuscript. See also reply to Reviewer 1, major comment 3.

3) Further regarding the hydrogen-deuterium exchange experiments, one would assume that direct

electrostatic disruption of the (810-830)-SAM(C-terminal half) interactions, if they indeed exist, would affect the hydrogen-deuterium exchanges much more than indirect disruption via mutations in the linker region; this, though, is not the case (compare Fig. 5A to Fig. 5F in the SAM region). In addition, the linker region mutations seem to similarly affect the hydrogen-deuterium exchanges in the N-terminal and C-terminal SAM regions (Fig. 5), which is not consistent with only the C-terminal region participating in the Kinase-SAM interactions (as suggested by Fig. 4).

Reply. While larger effects of direct electrostatic disruption are conceivable, it is difficult to precisely predict what the expected effects of the different mutations are, particularly since peptides derived from the kinase domain region of interest (residues 805-833) could not be analyzed in the HDX-MS experiments with the EphA2 E820K-E825K mutant (black bar in Fig. 5f and compare Fig. S6e with S6f).

With regard to the comment about hydrogen-deuterium exchange in the N-terminal and C-terminal SAM regions, it should be noted that the peptides derived from the SAM domain are rather long and many of the peptides that include the C-terminal helix (starting after Pro952), which we believe is one of the main regions of interaction with the kinase domain, also include regions much more N-terminal (many as N-terminal as K935, but some as far N-terminal as M926). This makes it difficult to pinpoint the precise region of the SAM domain that undergoes changes in hydrogen-deuterium exchange.

In our revised manuscript we now more clearly highlight an ‘ensemble view’ of the kinase and SAM domain conformations, meaning that different closed and open conformations that utilize different interaction surfaces may exist. This may explain why mutation of E820 and E825, which specifically disrupts a single interaction surface, has a relatively lower effect on the SAM domain than the 3E or 5E mutations, which more generally shift the EphA2 ensemble to more open conformations. We also note that while it is correct that the E820K/E825K mutations have smaller effects than the 3E and 5E mutations on the SAM domain, the effects of the E820K/E825K mutations on the juxtamembrane segment and activation loop are on par with the effects observed with the 5E mutation and stronger than those observed with the 3E mutation.

4) How do the authors explain the observed “stabilization of the kinase-SAM linker” in the mutations that are supposed to promote the “open” conformation (lines 262-265), when the intuitive expectation would be that an “open” conformation would have a less stable and more flexible linker?

Reply. While kinase and SAM domains do not interact in the open conformation, this does not preclude that the linker could be engaged in interactions with one of the domains rather than being fully flexible and disordered. And depending on their nature, such interactions are not necessarily incompatible with an open conformation. We have modified the text on page 6 of the Results to better explain this point.

5) If the authors claim that the (810-830)-SAM(C-terminal half) interactions are central to the formation of the “closed” conformation, why were mutations directly disrupting these electrostatic interactions (e.g. E820K/E825K) not tested in the FSI-FRET cell-based assays (lines 280-285)? Note that such interactions, if indeed existing, could possibly form both intra-molecularly and inter-molecularly on the cell surface.

Reply. The effects observed in the FRET experiments are small, which suggests that the negative charges of the linker do not have a strong effect on EphA2 dimerization in the absence of ligand. Therefore, we did not pursue this line of research with other mutants. However, we have now assessed the oligomerization of the 5E and 5A mutants in the presence of ligand, using a technique that is appropriate for oligomerization studies. This technique, Fluorescence Intensity Fluctuations (FIF) Spectrometry, is new and was not yet implemented in the Hristova lab at the

time of the original submission. The new data in Fig. 6 of the revised manuscript show that the negative charges in the linker region affect ligand-induced EphA2 oligomerization.

6) The authors use the terms “allosteric regulation” and “allosteric regulatory network” throughout the manuscript, title, discussion, and proposed model, but their data does not directly show any allostery in any of their experiments. The closest link to possible allosteric effects I could find is an observation of a small difference in the structure of a surface loop (lines 161-162, Fig. 2B) that has been previously proposed to have a possible role in an allosteric network in another receptor (EphA3)?

Reply. We have eliminated the references to allosteric regulation. See also the reply to the second general comment of Reviewer 1.

7) Lines 245-249: “Surprisingly, the effect of a single negative charge at position 897, the functionally best characterized linker phosphorylation site, does not produce a particularly strong effect on hydrogen-deuterium exchange in these regions (Fig. 5E), suggesting that phosphorylation of S897 alone is not sufficient to regulate EphA2 signaling properties.” Not really. What this more likely suggests is that the hydrogen-deuterium exchange changes in these regions may not adequately (or fully) represent the regulation of the EphA2 signaling properties via S897 phosphorylation.

Reply. We have modified the sentence on page 6 of the Results to “suggesting that phosphorylation of S897 alone is not sufficient to regulate EphA2 conformational changes that depend on linker negative charges”. In addition, however, on page 10 of the Discussion we mention the possibility that S897 phosphorylation may have effects through mechanisms other than contributing to cumulative negative charges in the linker.

Fig. 2 for the reviewers. The EphA2 linker can be simultaneously phosphorylated on 4 sites (S897, T898, S899 and S901). Evidence showing the 4 phosphosites is outlined in red (<http://141.61.102.18/phosida/ptm/eukaryotes/intermediate.aspx?species=homosapiens&query=IPI00745296&>).

Minor points:

8) The authors state that in vivo a maximum of 3 phosphorylated residues in the Kinase-SAM linker region are observed (line 91), yet in order to get significant effects in their biophysical studies they have to mutate all 5 potential phosphorylation sites. Is mutating all 5 sites reflective of a biologically relevant state?

Reply. We do not know for sure the maximum number of EphA2 linker residues that are concomitantly phosphorylated in cells (given that some phosphorylation could be lost during sample processing for mass spec). However, up to 4 phosphorylation sites have been detected by mass spectrometry (see Fig. 2 for the reviewers from ²² and <http://141.61.102.18/phosida/ptm/eukaryotes/intermediate.aspx?species=homosapiens&query=IPI00745296&>; the EphA2 phosphorylation sites can be found by clicking the phosphosite of interest on the left bar and then choosing “Kinome analysis”). In addition, it should be noted that a phosphate group is associated with a higher negative charge than glutamate, and that 3 phosphate groups approximately mimic 5 glutamic acids in terms of the negative charge introduced¹⁹. Thus, using the 5E mutant is not inconsistent with the mass spec data. We mention this on page 9 of the revised manuscript.

9) Lines 131-133, “The conformational flexibility of the S901 motif also supports the previously proposed idea that the negative charge introduced by S897 phosphorylation, and not a specific conformational change induced by S897 phosphorylation, is critical for subsequent phosphorylation of S901 by CK1 acidophilic kinases”. Not clear how it supports it?

Reply. We have eliminated this sentence.

References

1. Thorsness PE, Koshland DE, Jr. Inactivation of isocitrate dehydrogenase by phosphorylation is mediated by the negative charge of the phosphate. *J Biol Chem* **262**, 10422-10425 (1987).
2. Hiscott J, *et al.* Triggering the interferon response: the role of IRF-3 transcription factor. *J Interferon Cytokine Res* **19**, 1-13 (1999).
3. Martin I, *et al.* Ribosomal protein s15 phosphorylation mediates LRRK2 neurodegeneration in Parkinson's disease. *Cell* **157**, 472-485 (2014).
4. Feng Y, *et al.* Phosphomimetic mutants of pigment epithelium-derived factor with enhanced anti-choroidal melanoma cell activity in vitro and in vivo. *Invest Ophthalmol Vis Sci* **53**, 6793-6802 (2012).
5. Pearlman SM, Serber Z, Ferrell JE, Jr. A mechanism for the evolution of phosphorylation sites. *Cell* **147**, 934-946 (2011).
6. Iglesias T, Waldron RT, Rozengurt E. Identification of in vivo phosphorylation sites required for protein kinase D activation. *J Biol Chem* **273**, 27662-27667 (1998).
7. Kassenbrock CK, Anderson SM. Regulation of ubiquitin protein ligase activity in c-Cbl by phosphorylation-induced conformational change and constitutive activation by tyrosine to glutamate point mutations. *J Biol Chem* **279**, 28017-28027 (2004).
8. Ryan PE, Sivadasan-Nair N, Nau MM, Nicholas S, Lipkowitz S. The N terminus of Cbl-c regulates ubiquitin ligase activity by modulating affinity for the ubiquitin-conjugating enzyme. *J Biol Chem* **285**, 23687-23698 (2010).
9. Lai S, Pelech S. Regulatory roles of conserved phosphorylation sites in the activation T-loop of the MAP kinase ERK1. *Mol Biol Cell* **27**, 1040-1050 (2016).
10. Schweiger R, Linial M. Cooperativity within proximal phosphorylation sites is revealed from large-scale proteomics data. *Biology direct* **5**, 6 (2010).

11. Salazar C, Hofer T. Multisite protein phosphorylation--from molecular mechanisms to kinetic models. *FEBS J* **276**, 3177-3198 (2009).
12. Serber Z, Ferrell JE, Jr. Tuning bulk electrostatics to regulate protein function. *Cell* **128**, 441-444 (2007).
13. Holt LJ, Tuch BB, Villen J, Johnson AD, Gygi SP, Morgan DO. Global analysis of Cdk1 substrate phosphorylation sites provides insights into evolution. *Science* **325**, 1682-1686 (2009).
14. Ma B, Tsai CJ, Haliloglu T, Nussinov R. Dynamic allostery: linkers are not merely flexible. *Structure* **19**, 907-917 (2011).
15. Papaleo E, Saladino G, Lambrugh M, Lindorff-Larsen K, Gervasio FL, Nussinov R. The Role of Protein Loops and Linkers in Conformational Dynamics and Allostery. *Chemical reviews* **116**, 6391-6423 (2016).
16. Kellmann SJ, Dubel S, Thie H. A strategy to identify linker-based modules for the allosteric regulation of antibody-antigen binding affinities of different scFvs. *mAbs* **9**, 404-418 (2017).
17. Markwick PRL, Peacock RB, Komives EA. Accurate Prediction of Amide Exchange in the Fast Limit Reveals Thrombin Allostery. *Biophysical journal* **116**, 49-56 (2019).
18. Zisch AH, *et al.* Replacing two conserved tyrosines of the EphB2 receptor with glutamic acid prevents binding of SH2 domains without abrogating kinase activity and biological responses. *Oncogene* **19**, 177-187 (2000).
19. Dephoure N, Gould KL, Gygi SP, Kellogg DR. Mapping and analysis of phosphorylation sites: a quick guide for cell biologists. *Mol Biol Cell* **24**, 535-542 (2013).
20. Himanen JP, *et al.* Architecture of Eph receptor clusters. *Proc Natl Acad Sci U S A* **107**, 10860-10865 (2010).
21. Seiradake E, *et al.* Structurally encoded intraclass differences in EphA clusters drive distinct cell responses. *Nat Struct Mol Biol* **20**, 958-964 (2013).
22. Wissing J, *et al.* Proteomics analysis of protein kinases by target class-selective prefractionation and tandem mass spectrometry. *Mol Cell Proteomics* **6**, 537-547 (2007).

REVIEWERS' COMMENTS

Reviewer #1 (Remarks to the Author):

The reviewers have in general improved the interpretation in the revised manuscript.

Changes in the title and extensive changes in the amended discussion have improved the overall manuscript.

The abstract still needs to be modified:

- 1) The use of 'gradual' in gradual changes is vague! "We show that accumulation of multiple linker negative charges, mimicking phosphorylation, induces gradual changes in the EphA2 intracellular region from more closed to more extended conformations". Do you mean cooperative?
- 2) Replace 'cooperation' with 'coordination' in "Our findings suggest complex effects of linker phosphorylation on EphA2 signaling and imply that cooperation of multiple kinases is necessary to promote EphA2 noncanonical signaling"
- 3) In the same sentence above, 'complex' is once again unclear and needs to be replaced.

Reviewer #2 (Remarks to the Author):

The response to the comments raised by this reviewer #2 and also reviewer #1 are overall adequate. Additional SAXS experiments and a PISA analysis were done as requested. Overall several of the claims have been toned down.

Some sentences, including those newly introduced or revised are still suggesting overinterpretation and could be misleading. Specifically, in abstract "...induces gradual changes...and identify multiple kinases catalyzing linker phosphorylation". "gradual" is not an accurate description since little happens until 3E and 5E mutants. "and identify multiple kinases..." suggests that the two parts of this project are closely linked, i.e. more/more diverse kinases are engaged when the protein is in its open form. I am not sure the peptide data can support this directly. The confusion/inferred linkage may be avoided by making the "and identify" as separate sentence "We identify..."

The next sentence in the abstract also should be more specific, for its second part "Linker mutations promote EphA2 oligomerization in cells" -- again a direct linkage is being created and I am not sure about the evidence. "conformational changes in the JM segment and kinase domain" - unsure HDX supports conformational changes in these segments, rather the linker/SAM domain may occlude those segments in a configurational change, whereas JM and KD could be unchanged in their structure.

p.4 penultimate paragraph "may induce allosteric changes in the kinase domain" -- the authors said in their reply they had eliminated suggestions for allosteric change in the manuscript.

p.6 top - the PISA analysis should be referred to. Here, the crystallographic dimer (which the authors had essentially eliminated" is, nevertheless, introduced.

p.6 bottom "Thus phosphorylation of the N-terminal portion..initiates changes within the linker and has long-range conformational effects..." - this reads like allostery again, but also the certainty of this statements needs to be toned down- it is an inference from HDX data with mutants. Soften "A parsimonious interpretation of the data suggests the following mechanism.."

This reviewer notes that the FRET analysis reveals exciting differences between WT/5E and the 5A mutant, yet no HDX-MS or SAXS measurements are reported. Rather than enhancing the story, the single S892A and the 5A mutant seems to point to the limitations in comparing biophysical experiments with purified proteins and the in cell experiments. The interpretation is that dimerization is less because those residues can not be phosphorylated in cells, but the similarity between WT and any of the E mutant dimerization curves would call this into question, given that E is only a partial substitute for phosphorylation. Rather mutations to Ala could dramatically alter the behavior of the linker, in case of 5A turning into a helix? What is happening at the structural level clearly requires more experiments, something the authors should acknowledge.

The discussion section is well written and there is not the overinterpretation mentioned in the sections above. The last sentence "implications ..of other Eph receptors" could perhaps be expanded on or if it is just EphA1, this should be clarified.

Technical description of HDX-MS. The materials section says protein was held for 5 mins and then diluted into the D20 buffer. However, the protein sample is not well described before this step-- the reader can only assume it is in H2O buffer, but this needs to be clarified. There should be a brief mention that at pH 7.9 HDX is largely base catalyzed and thus additional negative charge may have a local as well as non-local effect of slowing the exchange. However, more Deuterium incorporation is observed and this effect is likely to be small.

Reviewer #3 (Remarks to the Author):

The revised manuscript addresses my concerns and suggestions and I support its publication in Nature Communications. I believe the reported results are noteworthy and significant to the field. Now, the revised interpretations and discussion match much more closely the experimental results.

Replies to the Reviewers' Comments

Reviewer #1

The reviewers have in general improved the interpretation in the revised manuscript. Changes in the title and extensive changes in the amended discussion have improved the overall manuscript.

The abstract still needs to be modified:

1) The use of 'gradual' in gradual changes is vague! "We show that accumulation of multiple linker negative charges, mimicking phosphorylation, induces gradual changes in the EphA2 intracellular region from more closed to more extended conformations". Do you mean cooperative?

Reply. We have replaced "gradual" with "cooperative" as recommended.

2) Replace 'cooperation' with 'coordination' in "Our findings suggest complex effects of linker phosphorylation on EphA2 signaling and imply that cooperation of multiple kinases is necessary to promote EphA2 non-canonical signaling"

Reply. We have replaced "cooperation" with "coordination" as recommended.

3) In the same sentence above, 'complex' is once again unclear and needs to be replaced.

Reply. We have replaced "complex" with "multiple". The multiple effects are described in the prior sentence.

Reviewer #2

The response to the comments raised by this reviewer #2 and also reviewer #1 are overall adequate. Additional SAXS experiments and a PISA analysis were done as requested. Overall several of the claims have been toned down.

Some sentences, including those newly introduced or revised are still suggesting overinterpretation and could be misleading. Specifically, in abstract "...induces gradual changes...and identify multiple kinases catalyzing linker phosphorylation". "gradual" is not an accurate description since little happens until 3E and 5E mutants. "and identify multiple kinases..." suggests that the two parts of this project are closely linked, i.e. more/more diverse kinases are engaged when the protein is in its open form. I am not sure the peptide data can support this directly. The confusion/inferred linkage may be avoided by making the "and identify" as separate sentence "We identify..."

Reply. We have replaced "gradual" with "cooperative" as recommended by Reviewer #1. We have split the sentence as recommended.

The next sentence in the abstract also should be more specific, for its second part "Linker mutations promote EphA2 oligomerization in cells" -- again a direct linkage is being created and I am not sure about the evidence. "conformational changes in the JM segment and kinase domain" - unsure HDX supports conformational changes in these segments, rather the linker/SAM domain may occlude those segments in a configurational change, whereas JM and KD could be unchanged in their structure.

Reply. We have changed this part of the abstract according to the reviewer's comments. However, we would like to keep "promote EphA2 oligomerization". It seems to us that if an EphA2 mutant shows increased oligomerization, then the mutation promotes oligomerization.

p.4 penultimate paragraph "may induce allosteric changes in the kinase domain" -- the authors said in their reply they had eliminated suggestions for allosteric change in the manuscript.

Reply. We have replaced “may induce allosteric changes in the kinase domain.” with “may induce conformational changes in the α FG loop of kinase domain.”

p.6 top - the PISA analysis should be referred to. Here, the crystallographic dimer (which the authors had essentially eliminated" is, nevertheless, introduced.

Reply. PISA analysis focuses on stable intermolecular interactions. We do not discuss an EphA2 homodimer in this section. Instead, we write that an interaction observed in the crystal structure may mimic intramolecular interactions between the kinase and SAM domains of a single EphA2 molecule. Hence, we do not believe that the PISA analysis is of much relevance in this context.

p.6 bottom "Thus phosphorylation of the N-terminal portion..initiates changes within the linker and has long-range conformational effects..." - this reads like allostery again, but also the certainty of this statements needs to be toned down- it is an inference from HDX data with mutants. Soften "A parsimonious interpretation of the data suggests the following mechanism.."

Reply. We have modified the text according to the reviewer recommendation.

This reviewer notes that the FRET analysis reveals exciting differences between WT/5E and the 5A mutant, yet no HDX-MS or SAXS measurements are reported. Rather than enhancing the story, the single S892A and the 5A mutant seems to point to the limitations in comparing biophysical experiments with purified proteins and the in cell experiments. The interpretation is that dimerization is less because those residues can not be phosphorylated in cells, but the similarity between WT and any of the E mutant dimerization curves would call this into question, given that E is only a partial substitute for phosphorylation. Rather mutations to Ala could dramatically alter the behavior of the linker, in case of 5A turning into a helix? What is happening at the structural level clearly requires more experiments, something the authors should acknowledge.

Reply. We have changed “imply” to “suggest” in the last sentence of the FRET section of the Results (Taken together, our FRET and FIF analyses suggest that kinase-SAM linker phosphorylation promotes EphA2 oligomerization in the plasma membrane.) and added the sentence “although further work is needed to study potential structural effects of multiple alanine mutations in the EphA2 linker” in the FRET section of the Discussion.

The discussion section is well written and there is not the overinterpretation mentioned in the sections above. The last sentence "implications ..of other Eph receptors" could perhaps be expanded on or if it is just EphA1, this should be clarified.

Reply. We have clarified that our study has implications for the functional regulation of EphA1 and structural understanding of other Eph receptors.

Technical description of HDX-MS. The materials section says protein was held for 5 mins and then diluted into the D2O buffer. However, the protein sample is not well described before this step-- the reader can only assume it is in H2O buffer, but this needs to be clarified. There should be a brief mention that at pH 7.9 HDX is largely base catalyzed and thus additional negative charge may have a local as well as non-local effect of slowing the exchange. However, more Deuterium incorporation is observed and this effect is likely to be small.

Reply. We now specify in the Methods section that the protein is in H₂O buffer before being diluted in D₂O buffer. To ensure comparison between the HDX-MS experiments of different EphA2 mutants, we performed all experiments under identical conditions and confirmed that deuteration nears a plateau within the time-course of our experiments (5 min). Therefore, the effects described by the reviewer are negligible in our analyses (as also acknowledged by the reviewer). In our view, this does not need to be further discussed/mentioned in the manuscript.

Reviewer #3

The revised manuscript addresses my concerns and suggestions and I support its publication in Nature Communications. I believe the reported results are noteworthy and significant to the field. Now, the revised interpretations and discussion match much more closely the experimental results.

Figure 8

Although not required by the reviewers, we have slightly modified the scheme in this figure to better illustrate our results.